

# Bound impurities in a one-dimensional Bose lattice gas: low-energy properties and quench-induced dynamics

Felipe Isaule[1⋆], Abel Rojo-Francàs[2,3] and Bruno Juliá-Díaz[2,3†]

**1** Instituto de Física, Pontificia Universidad Católica de Chile,
Avenida Vicuña Mackenna 4860, Santiago, Chile.
**2** Departament de Física Quàntica i Astrofísica, Facultat de Física,
Universitat de Barcelona, E-08028 Barcelona, Spain.
**3** Institut de Ciències del Cosmos, Universitat de Barcelona,
ICCUB, Martí i Franquès 1, E-08028 Barcelona, Spain.

⋆ felipe.isaule@uc.cl , † bruno@fqa.ub.edu

## Abstract

We study two mobile bosonic impurities immersed in a one-dimensional optical lattice and interacting with a bosonic bath. We employ the exact diagonalization method for small periodic lattices to study stationary properties and dynamics. We consider the branch of repulsive interactions that induce the formation of bound impurities, akin to the bipolaron problem. A comprehensive study of ground-state and low-energy properties is presented, including an examination of the interaction strengths which induce the formation of a bound dimer of impurities. We also study the dynamics induced after an interaction quench to examine the stability of the bound dimers. We reveal that after large interaction quenches from strong to weak interactions the system can show large oscillations over time with revivals of the dimer states. We find that the oscillations are driven by selected eigenstates with phase-separated configurations.

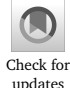
doi:10.21468/SciPostPhysCore.7.3.049

# 1  Introduction

The study of ultracold atomic mixtures has received increased attention in the past decades [1]. Mixtures have proved to show a plethora of interesting properties and phases, including non-dissipative drag [2,3], counterflow superfluidity [4,5], quantum droplets [6–10], among many others. Particularly interesting are mixtures with large population imbalances, where a minority species, usually referred to as impurities, is immersed in a majority species, which acts as a bath. The experimental progress realizing these highly-imbalanced gases has revitalized the interest in polaron physics [11]. Notably, impurities immersed in ultracold Bose gases, known as Bose polarons, were achieved in landmark experiments a few years ago [12–15].

One useful platform to study ultracold mixtures is one-dimensional optical lattices [16]. Theoretically, systems immersed in tight lattices can be described with a Hubbard-like model [17], which in one dimension can be solved with high numerical precision using exact diagonalization (ED) [18–20] and DMRG [21,22] techniques. Moreover, quantum effects become enhanced in one dimension [23,24], making one-dimensional systems good platforms to study novel effects in ultracold mixtures [25]. These advantages have motivated several recent studies of Bose-Bose [26–28] and Bose-Fermi [29–31] mixtures, as well as baths with single mobile impurities [32–35]. Experimentally, three-body losses are suppressed in one-dimensional gases [36], increasing their stability.

One fundamental configuration of highly-imbalanced mixtures consists of two mobile impurities immersed in a quantum bath. Two impurities often form bound particles usually referred to as bipolarons, which have received attention for many decades due to their connection with high-$T_c$ superconductivity [37,38]. More recently, and motivated by the progress realizing ultracold atomic mixtures, several theoretical studies have examined the bipolaron problem in Bose-Einstein condensates [39–47]. However, their experimental observation has remained elusive.

In tight one-dimensional optical lattices, ground-state properties of two impurities immersed in a bosonic bath have been studied with variational [48], DMRG [32] and ED [34] methods. Closely related, quench dynamics of impurities immersed in small lattices modeled with sinusoidal potentials have been studied in Ref. [49]. These works have shown that repulsive bath-impurity interactions induce the formation of bound impurity dimers, even in the absence of intra-impurity interactions. Furthermore, it has been shown that Mott-like baths induce the formation of tightly bound pairs [34,49].

In this work, we study the problem of two mobile bosonic impurities immersed in a tight one-dimensional lattice and interacting with a bosonic bath. We employ the ED method for small periodic lattices [19,20] and baths with unity filling. We study ground-state properties, including energy and average distances between atoms. This enables to provide a comprehensive characterization of the formation of dimers of impurities. In addition, we study the critical interactions where the two impurities tunnel together as a dimer. We also study the low-energy spectrum and examine the energy gaps. In particular, we analyze selected excited states in the limiting cases of small and strong interactions. Finally, we examine the time evolution of the system after a quench of the interactions. We analyze the evolution of the overlaps and average distance between atoms and characterize the periods of oscillations by examining the Fourier spectrum. We find that the system shows large oscillations within selected quench ranges and that these oscillations are mostly driven by eigenstates with phase-separated configurations.

This work is organized as follows. In Sec. 2 we detail our theoretical model and numerical approach. In Sec. 3 we examine ground-state properties, providing an exhaustive examination of the formation of dimers. In Sec. 4 we examine the low-energy spectrum of our model, to then in Sec. 5 study the dynamics after a quench of the interaction. Finally, we provide conclusions and an outlook for future directions in Sec. 6.

## 2 Model

We consider a tight one-dimensional optical lattice with $M$ sites loaded with a bath of $N_b = M$ bosons (unity filling) and two bosonic mobile impurities ($N_I = 2$). We model the system in consideration with a two-component Bose-Hubbard Hamiltonian, which assumes that all atoms are in the lowest Bloch band [17]. The Hamiltonian reads

$$\hat{H} = \hat{H}_{\text{hop}} + \hat{H}_{\text{int}} \, . \tag{1}$$

The first term describes the tunneling of atoms to the nearest neighbor sites

$$\hat{H}_{\text{hop}} = - \sum_{\sigma = b, I} J_\sigma \sum_i \left( \hat{a}^\dagger_{i,\sigma} \hat{a}_{i+1,\sigma} + \text{h.c.} \right) , \tag{2}$$

where $\hat{a}^\dagger_{i,\sigma}$ ($\hat{a}_{i,\sigma}$) creates (annihilates) a particle of species $\sigma$ at site $i$ and $J_\sigma > 0$ are the tunneling parameters. The interacting part describes the on-site interactions between atoms

$$\hat{H}_{\text{int}} = \frac{U_{bb}}{2} \sum_i \hat{n}_{i,b} \left( \hat{n}_{i,b} - 1 \right) + U_{bI} \sum_i \hat{n}_{i,b} \hat{n}_{i,I} \, , \tag{3}$$

where $\hat{n}_{i,\sigma} = \hat{a}^\dagger_{i,\sigma} \hat{a}_{i,\sigma}$ is the number operator and $U_{bb}$ and $U_{bI}$ are the strengths of the boson-boson and bath-impurity interactions, respectively. We consider that both interactions are repulsive ($U_{bb} \geq 0$ and $U_{bI} \geq 0$). Therefore, we focus on the repulsive branch of the impurity. We also stress that the impurities do not interact among themselves. This non-interacting limit is motivated by related studies of the bipolaron problem, where even if the impurities do not interact among themselves, the bath induces an effective impurity-impurity interaction which allows the bipolaron formation [39,45]. We illustrate the system in consideration in Fig. 1.

In the following, we consider that all species have the same tunneling parameters $J_b = J_I$. In contrast, we consider varying interaction strengths. In particular, we study the behavior of our model for a wide range of strengths $U_{bb}$ and $U_{bI}$.

Our model is inspired by the recent seminal experiments on two-component bosonic mixtures [7]. As mentioned, impurities can be achieved by inducing a large population imbalance between the two atomic species. In mixtures of the same atomic species, this can be achieved

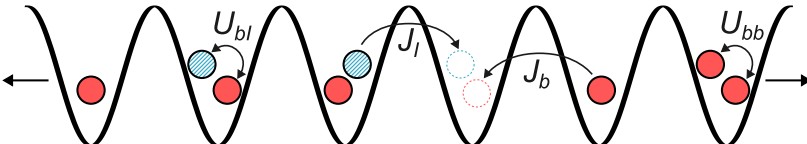

Figure 1: Illustration of the system. The solid red circles represent the bath's bosons, while the blue-hatched circles represent the impurities. The arrows on the edges of the figure represent the periodicity of the lattice.

by transferring atoms between hyperfine states with radiofrequency pulses [12]. We also note that we consider bath parameters ($U_{bb}/J_b$) which lie around the one-dimensional superfluid-to-Mott insulator (SF-MI) transition, as achieved in experiments [50], while the bath-impurity interaction can also be tuned with Feshbach resonances [51].

To study the system we employ the exact diagonalization (ED) method for a fixed number of particles [19,20] and consider lattices with six to nine sites and periodic boundary conditions (a ring). While ED restricts our calculations to lattices with a small number of sites, it enables us to easily study a wide range of properties with high precision, including excited states and dynamics. These advantages make ED a useful technique to study polaron properties [52]. Nevertheless, our setup is experimentally achievable, as ring configurations with a few sites can be produced experimentally with *ad-hoc* optical potentials [53,54]. In addition, the past decade has seen important progress in controlling systems with few atoms [25,55].

Within our ED approach, we employ the usual Fock basis in which each state is labeled by the occupations of the different sites of the lattice,

$$|\alpha\rangle = |n_{1,b}^{(\alpha)}, ..., n_{M,b}^{(\alpha)}; n_{1,I}^{(\alpha)}, ..., n_{M,I}^{(\alpha)}\rangle, \tag{4}$$

where $n_{i,\sigma}^{(\alpha)}$ represents the number of particles of species $\sigma$ in site $i$ for state $\alpha$. A wavefunction is then given by

$$|\Psi\rangle = \sum_\alpha c_\alpha |\alpha\rangle, \tag{5}$$

where the coefficients $c_\alpha$ are obtained from numerical diagonalization with the `ARPACK` library [56]. This library employs the iterative Lanczos algorithm, whose convergence is easy to control with well-chosen initial parameters, such as starting the Lanczos iteration from a previous solution.

We also provide analytical solutions in the limit of static atoms in App. A, i.e. when the tunneling is negligible compared to the interaction strengths. Finally, we stress that many of our results are specific to baths with unity filling. However, our conclusions can be generalized for more general bath configurations.

## 3 Ground-state properties

We start by examining the ground state of our Hamiltonian. Ground-state properties of similar models of one-dimensional lattices with bosonic baths and two impurities have been examined in previous related works, see for example Refs. [32,34,49,57]. Such works have shown that the bath induces the binding of the two mobile impurities into a dimer, which we refer to as a *di-impurity* bound state. These dimers are naturally characterized by a negative bipolaron energy, which acts as a binding energy and have sizes not much larger than the lattice spacing. In the following, we summarize some of these results but also provide a much more comprehensive

**SciPost**

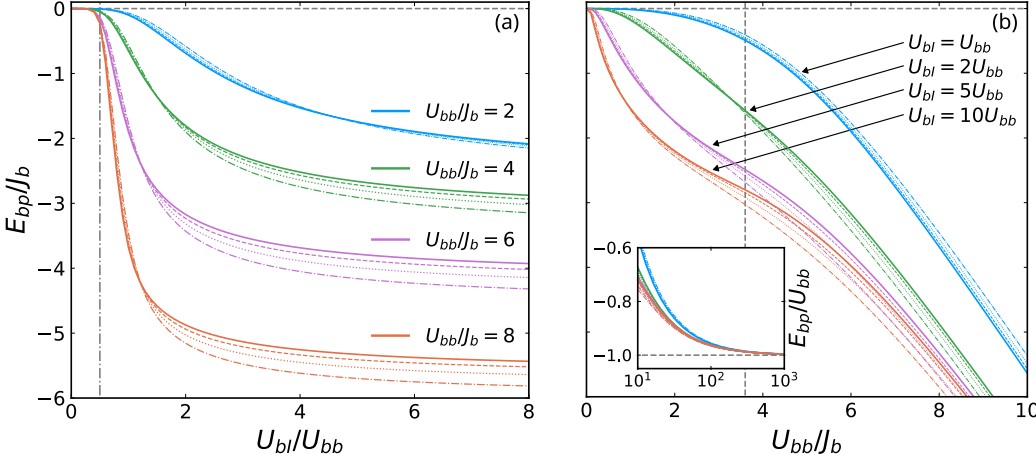

Figure 2: Bipolaron energy $E_{bp}$ as a function of $U_{bI}/U_{bb}$ (a) and of $U_{bb}/J_b$ (b). The curves with different colors indicate different bath parameters (a) and different bath-impurity interaction strengths (b) as indicated in the legends. The dash-dotted, dotted, dashed, and solid lines correspond to lattices with $M = 6, 7, 8, 9$ sites, respectively. The vertical dash-dotted line in (a) corresponds to $U_{bI} = U_{bb}/2$, while the vertical dashed line in (b) shows the SF-MI transition point for an infinite lattice [58]. The inset in (b) shows the behavior of $E_{bp}$ scaled by $U_{bb}$ for large $U_{bb}$.

examination of their properties. In particular, at the end of this section we examine the critical interaction strength for which the correlated tunneling of the two impurities is favored over the independent tunneling of single impurities. A complementary study of the ground state in the limit of static atoms is provided in A.1.

## 3.1 Binding energy

We first compute the binding energy of two impurities, or bipolaron energy, as reported previously in Refs. [32, 34]. It is defined as [39]

$$E_{bp}(U_{bb}, U_{bI}) = E_2(U_{bb}, U_{bI}) - 2E_1(U_{bb}, U_{bI}) + E_0(U_{bb}), \qquad (6)$$

where $E_2$ is the energy of the system with the two impurities, $E_1$ is the energy of the system with only one impurity, and $E_0$ is the energy of the system in the absence of impurities.

We note that the traditional picture of polarons is not fulfilled in optical lattices, as these do not support phonon excitations [16]. However, in the following, we refer to $E_{bp}$ as the bipolaron energy for consistency with related works (see for example Ref. [32]).

We show the bipolaron energy as a function of the bath-impurity interaction in Fig. 2(a). We show results for a representative set of bath parameters and different lattice sizes, as previously reported in Ref. [34]. We first stress that the bipolaron energy is negative in all cases, signaling the formation of bound states [32]. Moreover, and as previously shown in Ref. [34], the energies show a nice convergence with increasing lattice size $M$.

We observe that for weak repulsion $U_{bI} \lesssim U_{bb}/2$, $E_{bp}$ shows an approximate zero-energy plateau. This signals a miscible phase, where the bath is mostly undisturbed and the impurities move freely within the lattice without forming a bound state. On the other hand, for $U_{bI} \gtrsim U_{bb}/2$, the bipolaron energy decreases, which marks the formation of a bound di-impurity state. Moreover, $E_{bp}$ saturates to a finite value for large $U_{bI}$ [32]. As we show in more detail later, for $U_{bI} \gtrsim U_{bb}/2$ the system undergoes a *phase-separation* between the bath

and the impurities (immiscible phase) due to the strong bath-impurity repulsion. This induces the formation of a dimer bound state between impurities.

We note that $U_{bI} = U_{bb}/2$ corresponds to the critical interaction which separates the miscible and immiscible configurations in the static limit [see App. A]. Therefore, the mobile system naturally also shows a transition around $U_{bI} \approx U_{bb}/2$. However, we note that the small few-body and mobile systems considered in the main text show a *crossover* between a miscible and non-miscible phase instead of a well-defined transition. This crossover results in the smooth curves with continuous derivatives shown in Fig. 2(a). However, for large $U_{bb}$ the decrease around $U_{bI} \approx U_{bb}/2$ is more pronounced than for smaller $U_{bb}$, signaling a more abrupt crossover. The behavior for large $U_{bb}$ is examined in more detail in App. A.

We also show $E_{bp}$ as a function of $U_{bb}$ in Fig. 2(b) to examine the dependence of the bipolaron energy on the bath's parameters. We employ strongly-repulsive choices of $U_{bI}$ to study immiscible configurations. Panel (b) shows a distinct behavior of $E_{bp}$ for weak (left side of the panel) and strong (right side of the panel) bath's repulsion. For small $U_{bb}$ the bipolaron energy has a somewhat rich dependence on $U_{bb}$, showing a slow decrease for small $U_{bI}$, while showing a much rapid decrease for large $U_{bI}$. On the other hand, for large $U_{bb}$ the bipolaron energy depends linearly with $U_{bb}$ [34], diverging to $E_{bp} \to -\infty$ for $U_{bb} \to \infty$. Note that the inset shows that $E_{bp} \approx U_{bb}$ for very large $U_{bb}$. The latter corresponds to saturation to the static-limit solution $J_b = 0$ [see App. A.1]. However, we note that the saturation to the static limit is only achieved for very large $U_{bb}$ (see inset), which might be difficult to observe in experiment.

We also note that the change of behavior between weak and strong $U_{bb}$ occurs approximately around $U_{bb} \approx 4J_b$, near the superfluid-to-Mott transition point in an infinite lattice (vertical line). While we stress again that in our model we only observe a crossover and thus the SF-MI point is only a reference, we do observe a change of behavior between superfluid- and Mott-like baths. The latter will become clearer in the following.

## 3.2 Average distance between particles

To obtain a better physical picture of the behavior of the atoms and the formation of bound states, we examine the average distance between particles. In our ED formalism [see Eqs. (4) and (5)] it is obtained from

$$\langle r_{\sigma\sigma'} \rangle = \frac{d}{\mathcal{N}} \sum_{\alpha} \sum_{i,j} |c_\alpha|^2 n_{\sigma,i}^{(\alpha)} n_{\sigma',j}^{(\alpha)} r(i,j), \tag{7}$$

where $d$ is the lattice spacing,

$$r(i,j) = \min(|i-j|, M - |i-j|) \tag{8}$$

is the distance between two sites in a periodic lattice, and $\mathcal{N}$ is the number of distances to count. Between particles of different species $\sigma$ and $\sigma'$ we have that $\mathcal{N} = N_\sigma N_{\sigma'}$, while between equal species $\sigma$ we have $\mathcal{N} = \binom{N_\sigma}{2}$, where $\binom{\cdot}{\cdot}$ is the binomial coefficient.

Furthermore, to better compare the results for lattices with different numbers of sites, we scale the distances in terms of the average distance $r_0$ between two free bosons in a periodic one-dimensional lattice with $M$ sites

$$r_0 = \frac{d}{M} \sum_{i=2}^{M} [i/2], \tag{9}$$

where $[p/q]$ is the integer division between $p$ and $q$. Formula (9) can be obtained by considering all the possible ways to place two particles in a lattice with $M$ sites, associating a distance

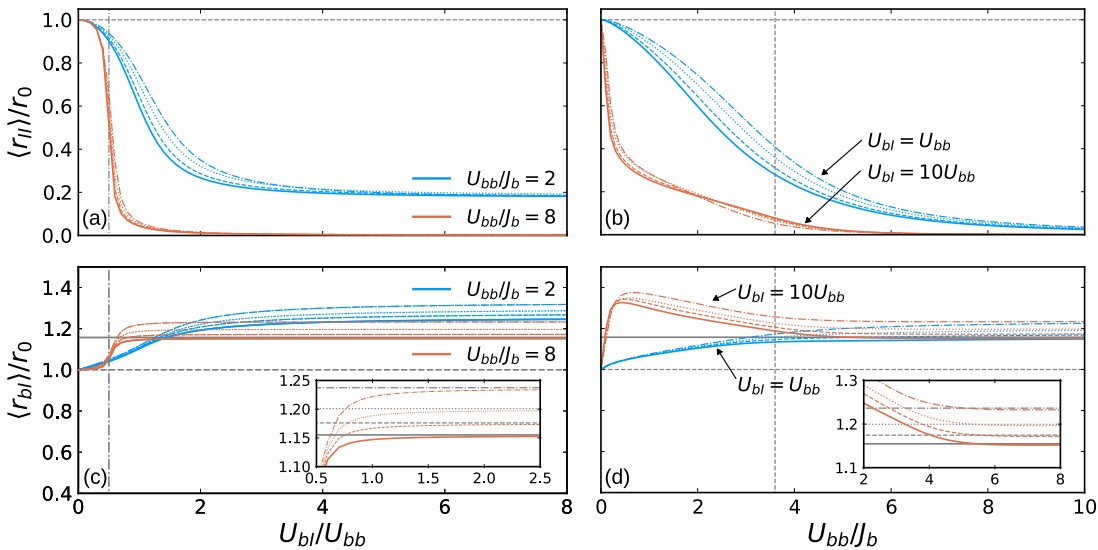

Figure 3: Average distance between the two impurities $\langle r_{II} \rangle$ (top panels) and between the bath and the impurities $\langle r_{bI} \rangle$ (bottom panels) as a function of $U_{bI}/U_{bb}$ (left panels) and of $U_{bb}/J_b$ (right panels). The curves with different colors indicate different bath parameters (left panels) and different bath-impurity interaction strengths (right panels) as indicated in the legends. The dash-dotted, dotted, dashed, and solid lines correspond to lattices with $M = 6, 7, 8, 9$ sites, respectively. The vertical dash-dotted lines in (a) and (c) correspond to $U_{bI} = U_{bb}/2$, while the vertical dashed lines in (b) and (d) show the SF-MI transition point for an infinite lattice [58]. The inset in (c) zooms a region for $U_{bb}/J_b = 8$ and in (d) a region for $U_{bI} = 10U_{bb}$. The horizontal lines in (c) and (d) correspond to $r_s^*$ for the corresponding lattice's size.

to each configuration, and then taking the average. We report values of $r_0$ for the lattice's sizes in consideration in Table 1.

We show the average distance between the two impurities $\langle r_{II} \rangle$ in the top panels of Fig. 3, which has previously been examined in Refs. [34, 49]. The distance $\langle r_{II} \rangle$ tells us the size of a di-impurity dimer if it is formed. In addition, we show the average distance between the bath's bosons and the impurities $\langle r_{bI} \rangle$ in the bottom panels to better illustrate the effect of the impurities on the bath. As with $E_{bp}$, the behavior of the average distances is similar between lattices with different sizes.

We first examine the behavior of the distances as a function of $U_{bI}$ (left panels). From panels (a) and (c), we first observe that for $U_{bI} \lesssim U_{bb}/2$ (left side of the panels) the aver-

Table 1: Average distance between two free bosons $r_0$ [Eq. (9)], between two free fermions of the same species $r_{F,0}$, and distance $r_s^*$ [Eq. (10)] for periodic one-dimensional lattices with $M$ sites.

| $M$ | $r_0/d$ | $r_{F,0}/d$ | $r_s^*/r_0$ |
|---|---|---|---|
| 6 | 1.5 | $\approx 2.17$ | $\approx 1.24$ |
| 7 | $\approx 1.71$ | $\approx 2.44$ | $\approx 1.2$ |
| 8 | 2.0 | $\approx 2.85$ | $\approx 1.18$ |
| 9 | $\approx 2.22$ | $\approx 3.14$ | $\approx 1.16$ |

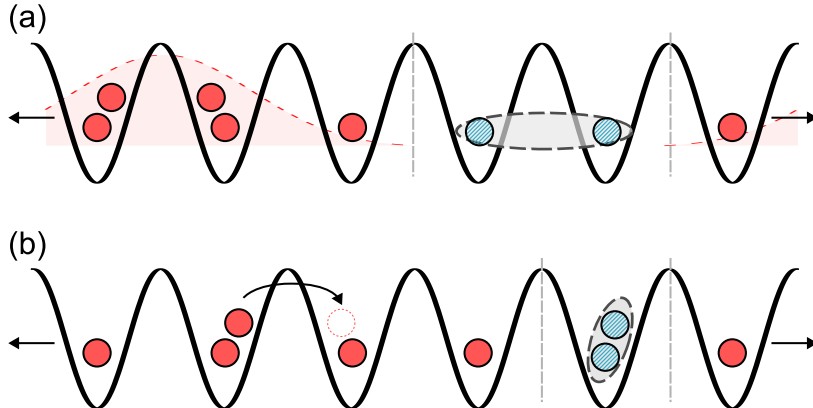

Figure 4: Illustration of the formation of the di-impurity dimers for $U_{bI} \gg U_{bb}$. The two impurities (blue-hatched circles) form a di-impurity bound state, which acts as a barrier to the bath's bosons (solid red circles). We illustrate the behavior for a superfluid-like bath in (a), and the behavior for a Mott-like bath in (b). The filled red Gaussian illustrates the superfluid bath. The vertical dashed lines illustrate the phase separation between the bath and the impurities.

age distances are roughly constant $\langle r_{\sigma\sigma'} \rangle \approx r_0$, signaling a miscible phase where the bath is mostly undisturbed and the impurities move freely. On the other hand, for increasing boson-impurity repulsion $U_{bI} > U_{bb}/2$ (right side of the panels) the distance between impurities $\langle r_{II} \rangle$ decreases, whereas $\langle r_{bI} \rangle$ increases. In both cases, the average distances saturate for large $U_{bI}$. We also again stress that we observe a smooth crossover around $U_{bI} \approx U_{bb}/2$, but this crossover is more abrupt for large $U_{bb}$ than more small $U_{bb}$.

From panel (a) we observe that $\langle r_{II} \rangle$ saturates to a value smaller than $r_0$ for large $U_{bI}/U_{bb}$. Note that $r_0$ is only slightly larger than the lattice spacing $d$ (see Table 1). Therefore, we can conclude that for large $U_{bI}/U_{bb}$ the two impurities are bound, forming a di-impurity dimer. In contrast, from panel (c) we observe that $\langle r_{bI} \rangle$ saturates to a value larger than $r_0$. Thus the distance between the bath and the impurities is larger than that with non-interacting impurities. This signals that the bath and the impurities undergo a phase separation (immiscible phase), which induces the formation of the di-impurity dimer separated from the bath.

In panel (a), for a weak bath's repulsion (blue lines) the di-impurity's size saturates to a small but finite value, while for a strong bath's repulsion (orange lines) the di-impurity's size vanishes. These results show that a weakly-repulsive bath in a superfluid-like state (small $U_{bb}$) induces shallow bound di-impurity dimers, while a strongly-repulsive bath in a Mott-like state (large $U_{bb}$) induces tightly bound dimers [34]. The reason for this behavior is that a di-impurity can compress a superfluid bath, and thus the dimer can expand, resulting in a large dimer. In contrast, a bath in a Mott-like state cannot be compressed. Therefore, due to the phase separation, such a bath occupies $M-1$ sites, while the two impurities are forced to occupy the one remaining site. The latter induces a tightly bound dimer with a vanishing size. We illustrate the formation of the di-impurity dimers in Fig. 4.

We also find that in the particular limit of $U_{bb} \gg J_b$ and $U_{bI} \gg U_{bb}$, the average distance $\langle r_{bI} \rangle$ saturates to [see inset in panel (b)]

$$r_s^* = r_0 + \frac{r_{F,0}}{M}, \tag{10}$$

where $r_0$ is given by Eq. (9) and $r_{F,0}$ is the average distance between two free fermions of the same species in a periodic lattice of $M$ sites. We report values of $r_{F,0}$ and $r_s^*$ in Table 1.

To understand the saturation of $\langle r_{bI} \rangle$ to $r_s^*$, we note again that due to the strong bath's bosons repulsion, $M - 1$ of the phase-separated bosons in the bath remain in a Mott-like state which occupy $M - 1$ sites. This accounts for the $r_0$ in Eq. (10). However, the remaining bath's boson can tunnel within the lattice [see Fig. 4(b)], but not occupy the same site as the impurities. The average distance between the impurities and this single mobile boson is captured by the fermionic distance in Eq. (10).

Following the previous examination, we can now better understand the behavior of the distances as a function of $U_{bb}$ (right panels of Fig. 3). Panel (b) nicely shows how $\langle r_{II} \rangle$ decreases for increasing $U_{bb}$, showing the smooth formation of smaller dimers. Note that while $\langle r_{II} \rangle$ shows a smoother decrease for smaller $U_{bI}$, the distance between impurities still vanishes for a sufficiently strong $U_{bb}$ if $U_{bI} > U_{bb}/2$.

Finally, Fig. 3(d) shows the increase and saturation of $\langle r_{bI} \rangle$ with increasing $U_{bb}$. In particular, the saturation of $\langle r_{bI} \rangle$ to $r_s^*$ for large repulsive interactions can also be appreciated in the inset, where the horizontal lines show $r_s^*$ for all the lattice sizes considered.

### 3.3 Dimer's tunneling

As explained before, the examined system shows a crossover from a miscible to phase-separation configuration around $U_{bI} \approx U_{bI}/2$ with no well-defined transition. The latter means that in principle we cannot find the precise interactions at which a di-impurity is formed. However, the formation of stable pairs can still be characterized more precisely by computing the tunneling correlator [59]

$$C_t = \langle a_{i,I}^\dagger a_{i,I}^\dagger a_{i+1,I} a_{i+1,I} \rangle - \langle a_{i,I}^\dagger a_{i+1,I} \rangle^2, \tag{11}$$

which compares the probability of two impurities tunneling together as a pair to that of the independent tunneling of single impurities. Indeed, the first term in Eq. (11) annihilates the two impurities at the same site and creates them at an adjacent one, while the second term corresponds to the tunneling of a single impurity. Note that in our periodic system the chosen site $i$ is arbitrary.

We show the correlator $C_t$ in Fig. 5 as a function of $U_{bI}$ [panel (a)] and as a function of $U_{bb}$ [panel (b)]. For a better comparison, we scale $C_t$ by its absolute value with non-interacting impurities

$$|C_t(U_{bI} = 0)| = \frac{2}{M^2}. \tag{12}$$

This expression is obtained by noting that two free bosons in a periodic lattice have that

$$\langle a_{i,I}^\dagger a_{i+1,I} \rangle^2 = \frac{2}{M}, \qquad \langle a_{i,I}^\dagger a_{i,I}^\dagger a_{i+1,I} a_{i+1,I} \rangle = \frac{2}{M^2}. \tag{13}$$

Panel (a) shows that for small $U_{bI}$ the correlator $C_t$ is negative, signaling that the two impurities do not tunnel together, as expected. This changes for $U_{bI} \gtrsim U_{bb}/2$, where $C_t$ becomes positive. The latter again signals the formation of di-impurity dimers, which tunnel as a pair within the lattice. We find that for Mott-like baths (orange lines) $C_t$ crosses zero almost exactly at $U_{bI} = U_{bb}/2$, consistent with the results for $E_{bp}$ and the average distances for large $U_{bb}$. In contrast, for weak bath repulsion $C_t$ becomes positive for slightly larger values of $U_{bI}$, showing that correlated dimers are formed for stronger bath-impurity repulsions.

Panel (b) nicely shows that for very large bath-impurity repulsion (orange lines) only a small bath's repulsion is needed to form a correlated dimer. On the other hand, for an intermediate value of $U_{bI}$ (blue lines), larger values of $U_{bb}$ are required.

We also note that both panels show that for strong inter-atomic repulsions (right side of both panels) $C_t$ vanishes, showing that it is not as favorable for the impurities to tunnel. This

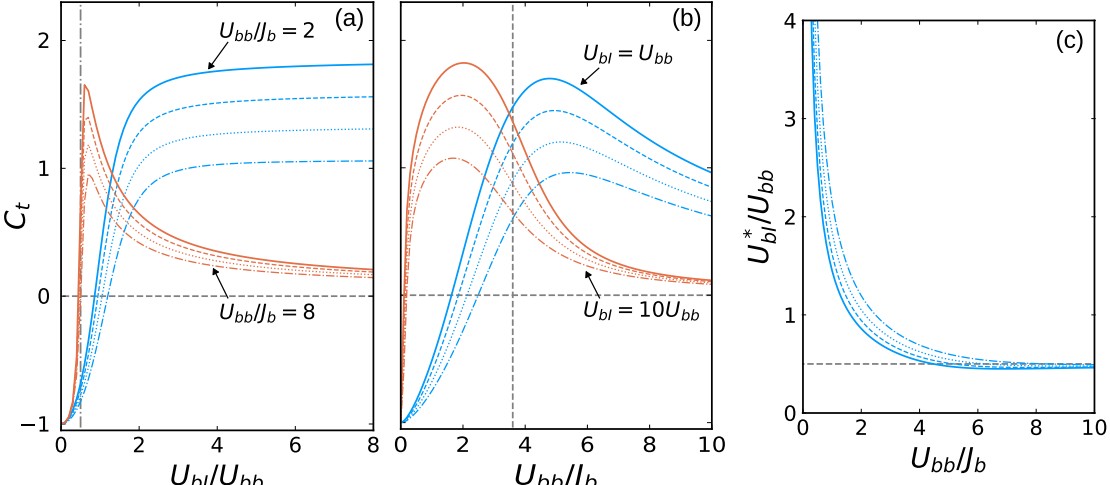

Figure 5: (a) and (b): Tunneling correlator $C_t$ [Eq. (11)] as a function of $U_{bI}/U_{bb}$ (a) and of $U_{bb}/J_b$ (b). The curves with different colors indicate different bath parameters (a) and different bath-impurity interaction strengths (b) as indicated in the legends. The vertical dash-dotted line in (a) corresponds to $U_{bI} = U_{bb}/2$, while the vertical dashed line in (b) shows the SF-MI transition point for an infinite lattice [58]. (c): Critical interaction strength $U_{bI}^*$ where $C_t = 0$ as a function of $U_{bb}$. The horizontal line in (c) corresponds to $U_{bI} = U_{bb}/2$. In all panels, the dash-dotted, dotted, dashed, and solid lines correspond to lattices with $M = 6, 7, 8, 9$ sites, respectively.

is expected when the tunneling parameter is small compared to the interaction strengths. Nevertheless, $C_t$ remains positive, showing that the dimer is still formed.

Finally, to quantify the exact point where correlated di-impurity dimers are formed, in Fig. 5(c) we show the critical interaction strength $U_{bI}^*$ where the tunneling correlator crosses zero $C_t = 0$. We can identify $U_{bI}^*$ as the critical interaction for the formation of correlated impurities. The panel also confirms that indeed for Mott-like baths with large $U_{bb}$ the critical point $U_{bI}^*$ saturates to

$$U_{bI,s}^* = U_{bb}/2 \,, \tag{14}$$

the exact phase-separation point in the static limit [see App. A].

On the other hand, as $U_{bb}$ decreases, the critical strength $U_{bI}^*$ increases. This is consistent with the smoother behavior shown by $E_{bp}$ and $\langle r_{\sigma\sigma'} \rangle$ for smaller values of $U_{bb}$.

# 4 Excited states

We now turn our attention to excited states. We first examine the energy gaps between the ground- and first-excited-state, to next study the low-energy spectrum. The latter enables us to better understand the dynamics, examined in Sec. 5.

## 4.1 Energy gaps

Here we examine the energy gap

$$\Delta E_2 = E_2^{(1)} - E_2^{(0)} \,, \tag{15}$$

where $E_2^{(0)}$ and $E_2^{(1)}$ correspond to the ground- and first-excited-state energy of the system with two impurities, respectively.

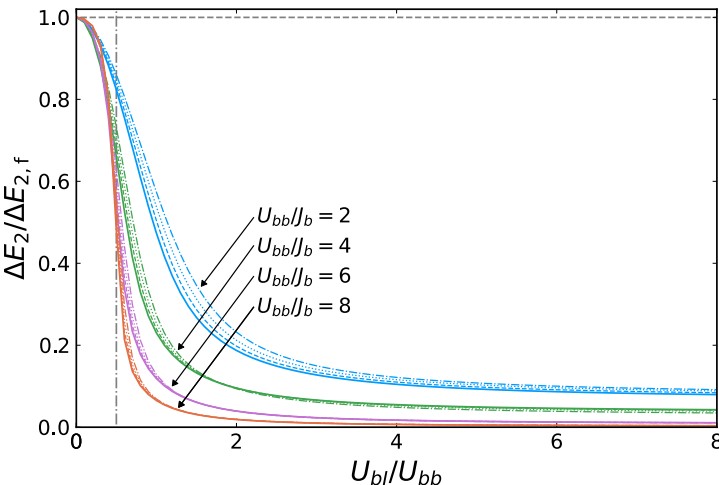

Figure 6: Energy gap $\Delta E_2$ between the ground and first-excited-state as a function of $U_{bI}/U_{bb}$. The curves with different colors indicate different bath-impurity interaction strengths, as indicated in the legends. The dash-dotted, dotted, dashed, and solid lines correspond to lattices with $M = 6, 7, 8, 9$ sites, respectively. The vertical dash-dotted line corresponds to $U_{bI} = U_{bb}/2$.

We show $\Delta E_2$ as a function of the bath-impurity interaction strength in Fig. 6. In addition, and to better compare lattices with different sizes, we scale the energies in terms of the energy gap $\Delta E_{2,f}$ for $U_{bI} = 0$. With this scaling we find that the gap behaves similarly for lattices with different sizes, showing a very weak dependence on the number of sites $M$.

First, we note that, for the chosen parameters, the energy gap for $U_{bI} = 0$ corresponds to that of a system with two free bosons. Therefore, at $U_{bI} = 0$ the first excitation simply corresponds to the excitation of the two free impurities, while the bath remains in its ground state. Therefore, we have that at $U_{bI} = 0$ [60],

$$\Delta E_{2,f} = -2J_I \left( \cos(2\pi/M) - 1 \right), \tag{16}$$

which is the known energy gap of free bosons in periodic lattices between the ground- and first-excited-states. Note that in our calculations we have employed $J_I = J_b$.

As $U_{bI}$ increases, the energy gap decreases, particularly around the previously discussed phase-separation crossover region $U_{bI} \approx U_{bb}/2$. We find that $\Delta E_2$ saturates for large $U_{bI}$. However, while for small $U_{bb}$ (blue and green lines) the gap saturates to a finite value, for large $U_{bb}$ (purple and orange lines) the gap closes. Indeed, we observe that for Mott-like baths the gap essentially vanishes for large $U_{bI}/U_{bb} \gg 4$, signaling that the ground state becomes degenerate for infinitely large $U_{bI}$. However, we stress that for the finite interaction strengths employed here, our numerical diagonalization still returns a very small but finite gap.

We stress that the ground state of our model for mobile atoms ($J_b > 0$) is non-degenerate, while the first excited state is two-fold degenerate. However, as just mentioned, in systems with Mott-like baths the ground state becomes almost degenerate for large bath-impurity repulsion, while it becomes $M$-times degenerate in the limit $U_{bI} \to \infty$. This is due to the $M$ sites available to have the tightly bound di-impurity dimers.

Interestingly, the energy gap decreases more abruptly around $U_{bI} \approx U_{bb}/2$ for larger values of $U_{bb}$. Once again, this is because in Mott-like baths the crossover between the miscible and phase-separation configuration is more abrupt, resulting in a rapid transition from a non-degenerate ground state to an almost degenerate one. We further examine this behavior in the following.

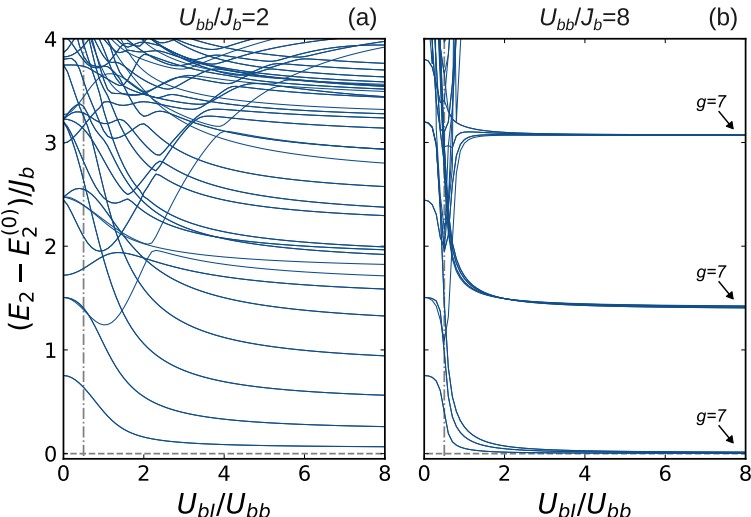

Figure 7: Low energy spectrum of a system with $M = 7$ sites as a function of $U_{bI}/U_{bb}$. The left panel (a) shows results for $U_{bb}/J_b = 2$ and the right panel (b) shows results for $U_{bb}/J_b = 8$. The vertical dash-dotted lines correspond to $U_{bI} = U_{bb}/2$. The selected legends indicate the degeneracy $g$ of the states.

## 4.2 Low-energy spectrum

We now examine the full low-energy spectrum $E_2$. We show the spectrum for a lattice with seven sites and as a function of the bath-impurity interaction in Fig. 7. We subtract the ground-state energy $E_2^{(0)}$ to better visualize the spectrum. We also stress that while the energies depend on the number of sites, the features discussed here are also present in lattices with different sizes, and thus we only show results for $M = 7$ for readability.

First, at $U_{bI} = 0$ we simply observe independent excitations of the uncoupled bath and impurities. In particular, for the periodic lattices considered here, we find excitations produced by persistent currents, which are given by [60]

$$E_{2,c}^{(n_b,n_I)} - E_2^{(0)} = -2 \sum_{\sigma=b,I} J_\sigma \left( \cos(2\pi n_\sigma/M) - 1 \right), \qquad (17)$$

where $n_\sigma = 0, 1, ..., M-1$ is the quantization of a circular current. Note that the gap Eq. (16) corresponds to $n_I = 1$ and $n_b = 0$.

For strong bath's repulsion [panel (b)], it is easy to see that all the shown excitation gaps at $U_{bI} = 0$ are nicely given by Eq. (17). This is expected due to the large value of $U_{bb}$. In contrast, for weak bath's repulsion [panel (a)], while some energy gaps are given by (17), the spectrum shows many additional intermediate bath's excitations.

For finite values of $U_{bI}$, the spectrum shows a very distinct behavior depending on the bath's repulsion. For large $U_{bb}$ [panel (b)], within the miscible phase ($U_{bI} \lesssim U_{bb}/2$) the excitation spectrum is approximately given by Eq. (17), as discussed. Then, around $U_{bI} \approx U_{bb}/2$, the states quickly deviate from their values for $U_{bI} \approx 0$, to then saturate for $U_{bI} \gg U_{bb}$ and become degenerate with other states [right side of panel (b)]. We find that these saturated states have a degeneracy of $M$, as indicated in the panel.

One striking feature shown at large $U_{bb}$ and $U_{bI}$ [right side of the panel (b)], is that the spectrum shows almost equally spaced excitations. We have found that these excited states for large $U_{bI}$ in panel (b) correspond to excitations of $M$ bosons in a lattice with $M-1$ sites and open boundary conditions. Indeed, if one solves a one-component Bose-Hubbard model

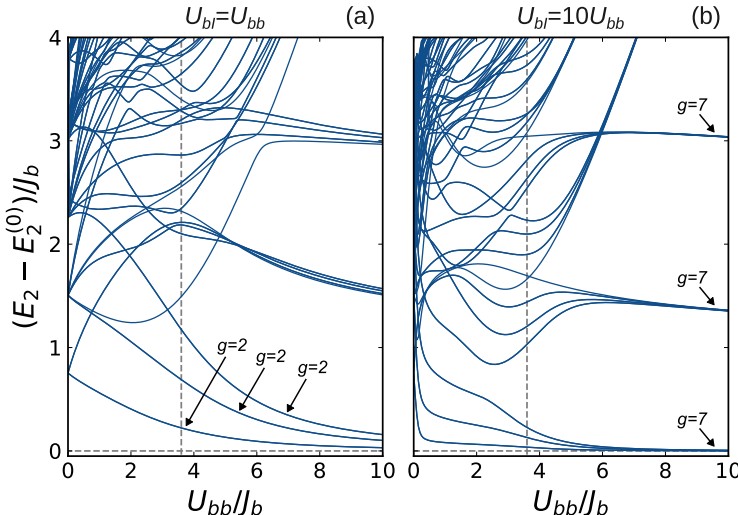

Figure 8: Low energy spectrum of a system with $M = 7$ sites as a function of $U_{bb}/J_b$. The left panel (a) shows results for $U_{bI} = U_{bb}$ and the right panel (b) shows results for $U_{bI} = 10U_{bb}$. The selected legends indicate the degeneracy $g$ of the states. The vertical dashed line shows the SF-MI transition point for an infinite lattice [58].

with $M$ bosons and $M - 1$ sites with open boundary conditions, one obtains the same energy gaps. Therefore, we can conclude that for large $U_{bI}$, the low excited states also show phase separation and formation of di-impurity dimers, and thus the impurities act as a barrier for the bath [see Fig. 4(b)]. The phase separation produces low energy excitations which correspond to those of the bath confined in the remaining $M - 1$ sites, while the dimer remains at its ground state. The degeneracy of $M$ can then be understood from the $M$ possible sites that the di-impurity dimers can occupy. However, here we stress that higher energy excitations (not shown in the panel) correspond to other excited configurations, even miscible ones which can break the di-impurity states.

On the other hand, for weak bath's repulsion [panel (a)], the spectrum shows a much richer behavior. This can be expected. Because the tunneling and interaction strengths have similar magnitudes, there is strong competition between these quantities, producing many crossings. In addition, while the energy gaps do saturate for large values of $U_{bI}$, these do not become degenerate with other states as in panel (b). Therefore, the states in panel (a) are either non-degenerate or two-fold degenerate. Moreover, the many excited states shown for large $U_{bI}$ [right side of the panel (a)] correspond to multiple different configurations, including excitations of either the bath or the impurities in phase-separated states, which will become relevant in the next section.

We also show the low-energy spectrum as a function of the bath's repulsion in Fig. 8. This figure nicely shows the transition between a rich spectrum for weak baths repulsion (left sides of the panels), to a simpler spectrum with highly-degenerate states for strong baths repulsion (right sides of the panels).

In both panels, for large $U_{bb}$ the low-energy states correspond to excitations of the bath confined in $M - 1$ sites, as previously explained. We have found that beyond the SF-MI crossover region $U_{bb} \approx 4J_b$, the spectrum saturates to these states. However, we note that for stronger bath-impurity repulsion the states become degenerate for much smaller values of $U_{bb}$.

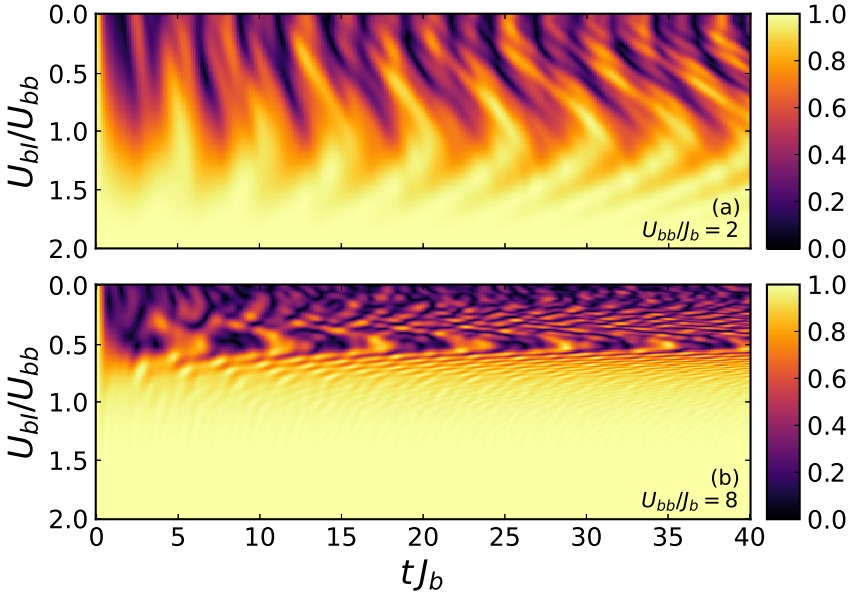

Figure 9: Overlap $|\langle\Psi_0|\Psi(t)\rangle|$ as a function of time $t$ for $M = 7$. The initial state $|\Psi_0\rangle$ is prepared in the ground state with $U_{bI} = 2U_{bb}$ (both panels), and with $U_{bb}/J_b = 2$ (a) and $U_{bb}/J_b = 8$ (b). An interaction quench in $U_{bI}$ is performed at $t = 0$ to the values indicated in the $y$-axis.

# 5 Quench dynamics

Having discussed stationary properties, we now examine the dynamics of the system induced by a quench of the interactions. We note that interaction quenches can be introduced with Feshbach resonances techniques [51].

In the following, we prepare an initial state $|\Psi_0\rangle$ in the ground state for finite interaction strengths $U_{bb}$ and $U_{bI}$. We then perform a sudden quench of the strength of either interaction at a time $t = 0$ to a lower value and solve the time evolution by numerical exponentiation of the Hamiltonian [61]

$$|\Psi(t)\rangle = e^{-i\hat{H}t}|\Psi_0\rangle. \tag{18}$$

The interaction quenches enable us to study oscillations of the di-impurity dimers, which could be probed experimentally by observing correlations between atoms, as done with other atomic mixtures [62, 63].

In the following, we examine the evolution of the overlaps and average distance between particles, as well as the periods of oscillations. To connect the oscillations with the spectra shown in Sec. 4, we only show dynamics for $M = 7$. We stress that we have checked that lattices with other sizes show similar results for the dynamics, and only the periods of oscillations depend on $M$. We also provide an examination of correlations in App. B.

## 5.1 Overlaps

We first examine the overlap between the initial state $|\Psi_0\rangle$ and the evolved state $|\Psi(t)\rangle$. In Fig. 9 we show the time evolution of the overlap for interaction quenches in $U_{bI}$. The interaction $U_{bb}$ remains fixed. In both panels, we prepare the initial state at a strong bath-impurity interaction $U_{bI} = 2U_{bb}$ so the impurities form a dimer. However, in panel (a) we choose a weak bath's repulsion with a shallow initial dimer, while in panel (b) we choose a strong bath's repulsion with a tightly bound dimer.

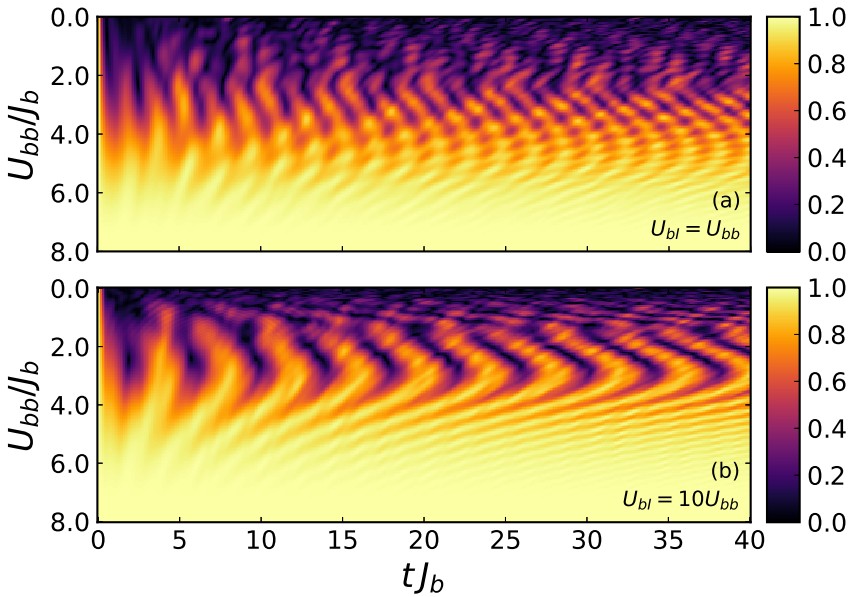

Figure 10: Overlap $|\langle \Psi_0 | \Psi(t) \rangle|$ as a function of time $t$ for $M = 7$. The initial state $|\Psi_0\rangle$ is prepared in the ground state with $U_{bb}/J_b = 8$ (both panels), and with $U_{bI} = U_{bb}$ (a) and $U_{bI} = 10U_{bb}$ (b). An interaction quench in $U_{bb}$ is performed at $t = 0$ to the values indicated in the $y$-axis.

We find that for small quenches (bottom sides of the panels), the overlap remains approximately constant and close to one. This is expected, as those small interaction quenches should not change the initial state significantly. In contrast, for larger quenches beyond the critical strength for correlated dimers [Fig. 5(c)] the overlaps show strong oscillations. For weak bath's repulsion [panel (a)] the overlap shows a smooth onset of oscillations, with periodic vanishing overlaps for $U_{bI} \lesssim U_{bb}$ (note that in Fig. 5(c) $U_{bI}^* \approx U_{bb}$ for $U_{bb} = 2J_b$). Nevertheless, the overlaps periodically return to values close to one for all examined quenches in $U_{bI}$, suggesting revivals of the initial state. On the other hand, for strong bath's repulsion [panel (b)], the overlap shows an abrupt onset of large oscillations around $U_{bI} \approx U_{bb}/2$ (again note that in Fig. 5(c) $U_{bI}^* \approx U_{bb}/2$ for $U_{bb} = 8J_b$). In addition, for larger quenches $U_{bI} < U_{bb}/2$, the overlap remains always small, with no visible revival of the initial state [upper region of the panel (b)]. This suggests the onset of an orthogonality catastrophe in Mott-like baths, as manifested in related systems of impurities [64, 65], and which will be examined in more detail in future work.

Interestingly, the oscillations increase their periods with increasing quench, reaching a maximum period for a finite value of post-quench $U_{bI}$. In panel (a) this maximum is reached at $U_{bI} \approx U_{bb}$, while in panel (b) is reached at $U_{bI} \approx U_{bb}/2$. Both correspond approximately to the the point where $C_t = 0$ [Fig. 5(c)]. A similar behavior has also been reported in related studies of impurities trapped instead in a one-dimensional harmonic trap [44, 64]. We examine this in more detail later in this section.

We also show overlaps for an interaction quench in $U_{bb}$ in Fig. 10. In this figure, the ratio $U_{bI}/U_{bb}$ remains fixed, so $U_{bI}$ is also changed. In both panels, the initial state is prepared with a Mott-like bath. However, both panels consider distinct bath-impurity ratios $U_{bI}/U_{bb}$.

As with the previous case, a small quench (bottom sides of the panels) does not change the initial state significantly, so the overlap is approximately one. However, a larger quench produces strong oscillations. We find that in both cases the onset of large oscillations occurs around the Mott-to-superfluid transition region $U_{bb} \approx 4J_b$ which, as discussed, separates the

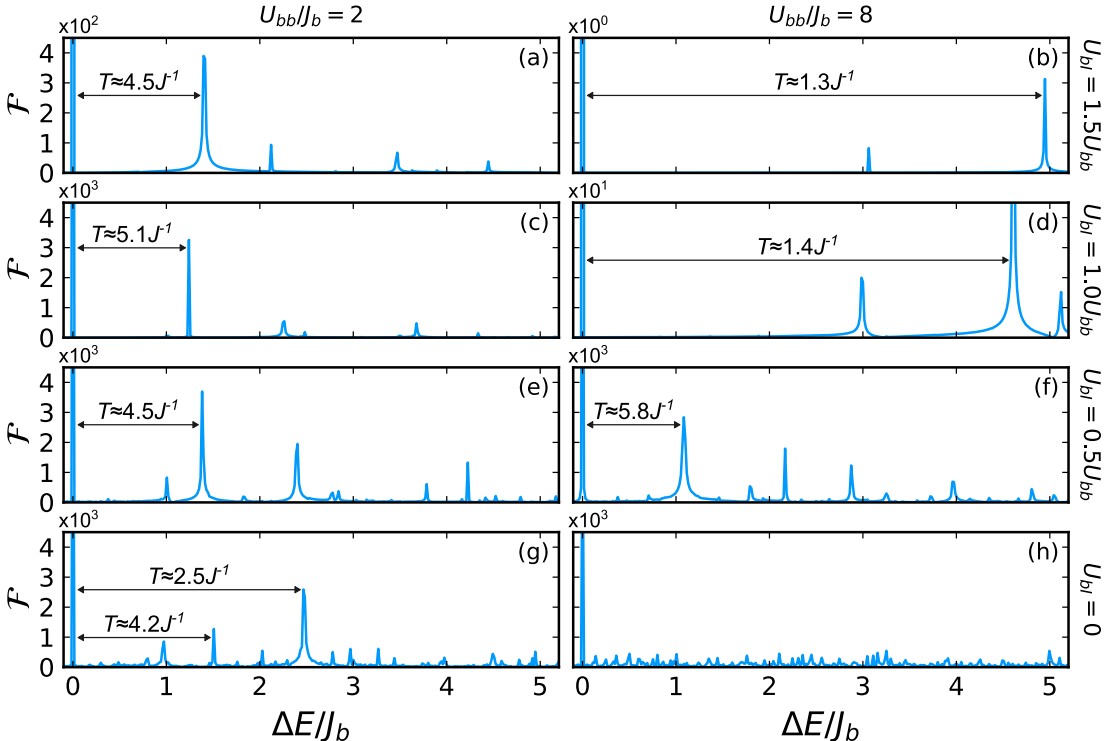

Figure 11: Fourier transform of the time-evolution of the overlap $|\langle\Psi_0|\Psi(t)\rangle|$ as a function of energy gaps $\Delta E = 2\pi\nu$. The initial state $\Psi_0$ is prepared in the ground state with $U_{bb}/J_b = 2$ (left panels) and $U_{bb}/J_b = 8$ (right panels). An interaction quench in $U_{bI}$ is performed at $t = 0$ to $U_{bI} = 1.5U_{bb}$ [panels (a) and (b)], $U_{bI} = 1.0U_{bb}$ [panels (c) and (d)], $U_{bI} = 0.5U_{bb}$ [panels (e) and (f)], and $U_{bI} = 0$ [panel (g) and (h)]. The time-evolution is performed up to a time $t = 40000J_b^{-1}$. The horizontal lines with arrows indicate the corresponding periods of oscillation.

regimes of small and large $U_{bb}$. We remind the reader that for $U_{bb} \lesssim 4J_b$ the impurities form shallow dimers, while for larger values of $U_{bb}$ the impurities form smaller tightly-bound dimers. Therefore, the onset of large oscillations in the overlap reflects the relaxation of the dimers. Interestingly, in both panels, we observe vanishing overlaps for quenches to $U_{bb} \approx 0$.

The quench in $U_{bb}$ also produces a change in the periods of oscillations, which reach a maximum for a finite value of the post-quench interaction $U_{bb}$. This is particularly notorious in panel (b), in which the period reaches a maximum around $U_{bb} \approx 3J_b$. We examine the periods of oscillations in the following.

## 5.2 Fourier analysis

To understand the oscillation patterns after the interaction quenches, we perform a Fourier analysis of the time evolution of the overlaps. We show the Fourier spectrum for a quench in $U_{bI}$ (to be compared with Fig. 9) in Fig. 11. We show analyses for selected bath-impurity interactions $U_{bI}$, and for a weak and strong bath's repulsion in the left and right panels, respectively. Note that in the $x$-axes we show energy gaps $\Delta E = 2\pi\nu$, where $\nu$ are the frequencies obtained from the Fourier transform. This enables us to better compare with the energy spectrum shown in Fig. 7.

The peaks correspond to the states with the largest overlaps with the initial state. Because the initial state corresponds to a phase-separated configuration, we have found that the peaks

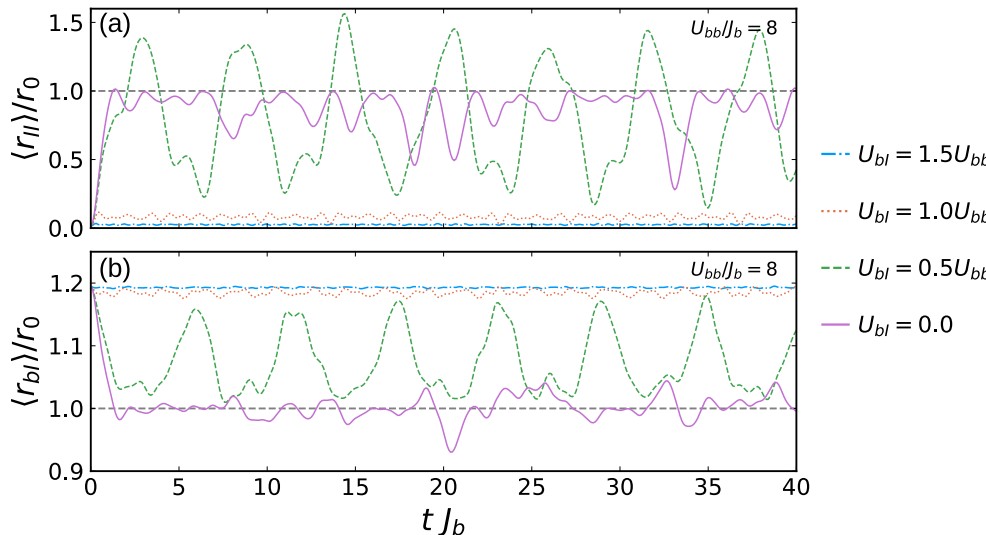

Figure 12: Average distance between impurities $\langle r_{II} \rangle$ (a) and between bath and impurities $\langle r_{bI} \rangle$ (b) as a function of time $t$ for $M = 7$. The initial state is prepared in the ground state with $U_{bI} = 2U_{bb}$ and $U_{bb}/J_b = 8$. An interaction quench in $U_{bI}$ is performed at $t = 0$ to the values indicated in the legends.

precisely correspond to phase-separated excitations. Interestingly, in panels (b) and (d), due to the strong repulsive interactions, the high-energy peaks at $\Delta E \approx 5J_b$ [beyond the range shown in Fig. 7(b)] correspond to phase-separated excitations where the two impurities occupy adjacent sites, and the bath occupies the remaining $M - 2$ sites. Nevertheless, note that such peaks are small, resulting in the small oscillations with short periods observed in Fig. 9(b).

In the figure, we also show the corresponding periods $T = \nu^{-1}$ between the peaks. It is easy to see that these periods agree with the oscillatory behavior shown in Fig. 9. Indeed, the periods increase for increasing quench, reaching a maximum at $U_{bI} \approx U_{bb}$ for $U_{bb}/J_b = 2$, and at $U_{bI} \approx U_{bb}/2$ for $U_{bb}/J_b = 8$.

Concerning the quenches to the limit of non-interacting impurities (bottom panels), we find that for $U_{bb}/J_b = 2$ [panel (g)] the Fourier analysis shows well-defined peaks, consistent with the oscillatory behavior found at $U_{bI} = 0$ in Fig. 9(a). Moreover, the Fourier analysis shows two large peaks with periods that are consistent with the two clear overlapping oscillations shown in 9(a). In contrast, for $U_{bb}/J_b = 8$, the Fourier analysis shows no clear frequencies, explaining the erratic behavior of the time-evolved overlap in Fig. 9(b) for $U_{bI} \to 0$.

## 5.3 Distances between particles

Having examined the time evolution of the overlaps, we now turn our attention to the evolution of the average distance between impurities. This enables us to provide a better physical picture of the behavior of the impurities after an interaction quench.

We show the time evolution of average distances between atoms for quenches in $U_{bI}$ in Fig. 12. We only show results for a strongly repulsive bath with $U_{bb}/J_b = 8$, as its evolution better conveys our results.

The average distance between impurities [panel (a)] shows small oscillations for small quenches (blue dash-dotted and orange dotted lines), with $\langle r_{II}(t) \rangle \approx 0$. This means that the di-impurity dimer remains tightly bound in the same single site. On the other hand, for large quenches that cross the interaction $U_{bI}^*$ where $C_t = 0$ (green dashed and violet solid lines),

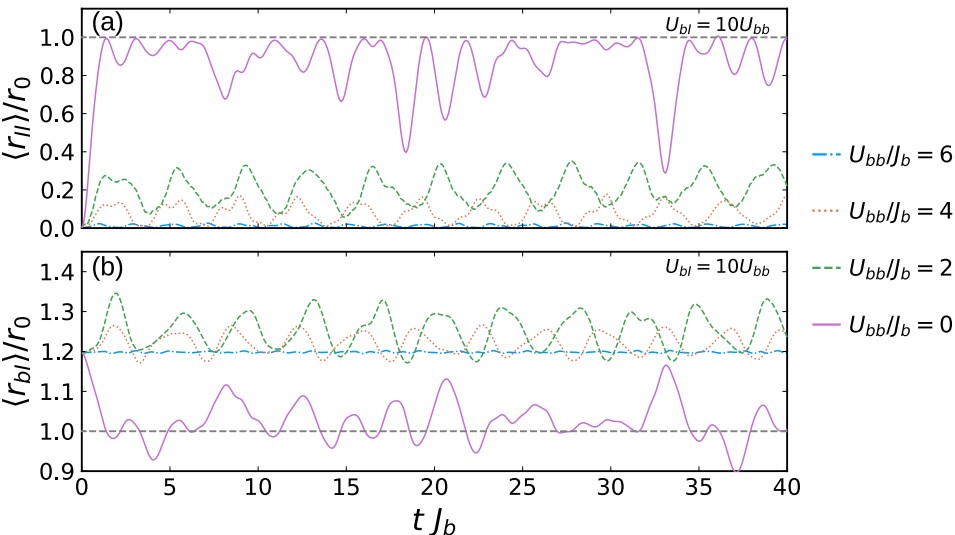

Figure 13: Average distance between impurities $\langle r_{II} \rangle$ (a) and between bath and impurities $\langle r_{bI} \rangle$ (b) as a function of time $t$ for $M = 7$. The initial state is prepared in the ground state with $U_{bb}/J_b = 8$ and $U_{bI} = 10U_{bb}$. An interaction quench in $U_{bb}$ is performed at $t = 0$ to the values indicated in the legends.

$\langle r_{II} \rangle$ shows large oscillations. Note that the onset of oscillations is consistent with that shown by the overlaps [see Fig. 9(b)]. Interestingly, a large quench to an interaction around the critical strength $U_{bI,s}^* = U_{bb}/2$ (green dashed line) produces oscillations around $r_0$ which seem to destroy and revive the dimer. In contrast, a quench to non-interacting impurities $U_{bI} = 0$ (solid lines) results in smaller oscillations with distances just below $r_0$. Therefore, in the latter case, the dimer is essentially destroyed with only partial revivals

The evolution of $\langle r_{bI} \rangle$ complements that of $\langle r_{II} \rangle$. Indeed, in general $\langle r_{bI} \rangle$ decreases when $\langle r_{II} \rangle$ increases, and vice-versa. For small interaction quenches $\langle r_{bI} \rangle$ shows only small oscillations around the saturation distance $\langle r_{bI} \rangle \approx 1.2r_0$. In contrast, a large quench beyond $U_{bI}^*$ produces sudden large oscillations. In particular, a quench to zero bath-impurity interaction $U_{bI} = 0$ (solid violet line) produces oscillations of $\langle r_{bI} \rangle$ around $r_0$, and thus the bath recovers its behavior when it is free of impurities.

Furthermore, we show the time evolution of average distances for quenches in $U_{bb}$ in Fig. 13. We show results for a very strong bath-impurity repulsion $U_{bI} = 10U_{bb}$. In this figure the interaction quenches to finite interactions maintain the dimers bound, but change their sizes. A quench to a smaller but still large bath's repulsion (blue dash-dotted line) maintains $\langle r_{II} \rangle$ and $\langle r_{bI} \rangle$ essentially unchanged. On the other hand, for quenches to a weak bath's repulsion (orange dotted and green dashed lines), the dimers remain bound, but their size increases. The latter results in larger oscillations of the average distances.

Finally, a quench to a completely non-interacting system (purple solid lines) results in a big change in the average distance. As expected, $\langle r_{II} \rangle$ quickly increases to $\approx r_0$ and then oscillates with values $\langle r_{II} \rangle < r_0$. Similarly, $\langle r_{bI} \rangle$ quickly starts showing oscillations around $r_0$, as expected.

To end this section, we again stress that lattices with different sizes show analogous oscillations with collapses and revivals of the di-impurity states. The main difference is that the periods of oscillations increase with increasing $M$. While this makes predictions for large lattices difficult, the revivals should be present in small ring geometries [53,54].

# 6 Conclusions

In this work, we provided a comprehensive study of stationary and quench-dynamics properties of the problem of two bosonic impurities immersed in a bosonic bath with unity filling in a tight one-dimensional optical lattice. We employed the exact diagonalization method and considered small lattices with different sizes. We found that the studied properties show a weak dependence on the lattice's size.

In the ground state, we confirmed the formation of bound di-impurity dimers induced by the phase separation between the bath and the impurities. Furthermore, we studied the correlated tunneling of the two impurities, which enabled us to estimate the interaction strengths that support the formation of dimers. We have found that di-impurity dimers essentially form for interactions $U_{bI} > U_{bb}/2$ in Mott-like baths, whereas superfluid-like baths show a smooth crossover between a miscible and a phase-separated configuration with bound dimers.

By examining excited states, we found that the energy spectrum for weak interactions shows a rich behavior due to the competition between the different parameters. On the other hand, when all the interactions are strongly repulsive, the first low-energy excitations correspond to those of a confined bath in an open lattice due to the phase-separated impurities.

Finally, by performing quenches from large interaction strengths to weaker interactions, we examined oscillations of the di-impurity dimers. We found that for large quenches beyond the ground-state characteristic strengths, the system shows large oscillations that can destroy and revive the dimer states. We also found that the oscillations are driven by excitations to phase-separated configurations.

The studied model could be realized experimentally with highly imbalanced bosonic mixtures confined in ring geometries, while the properties shown in this work could be probed with measurements of spin correlations. Such studies could provide important insight into related problems of bound bipolarons in quantum mediums and pairing phenomena in imbalanced atomic mixtures.

In the future, we intend to study larger lattices by employing other techniques, such as DMRG, in part to examine which of the studied features persist in larger many-body configurations. Moreover, the study of impurities immersed in bosonic baths across the superfluid-to-Mott transition is of particular interest, as it could complement recent studies in two-dimensional lattices [66–69]. The study of non-zero impurity-impurity interactions is also of interest, particularly the study of its competition with the bath's induced effective interaction. Other possible extensions include a further examination of excited states and of persistent currents, the consideration of dipolar [70–72] or distinguishable [73, 74] impurities, as well as the study of multiple impurities [75], which could provide insight into the study of imbalanced mixtures in optical lattices [76].

# Acknowledgements

We thank J. Martorell for the useful discussions and for carefully reading our manuscript.

**Funding information** F.I. acknowledges funding from ANID through FONDECYT Postdoctorado No. 3230023. B.J-D and A.R-F acknowledge funding from Grant No. PID2020-114626GB-I00 by MCIN/AEI/10.13039/5011 00011033 and "Unit of Excellence María de Maeztu 2020-2023" award to the Institute of Cosmos Sciences, Grant CEX2019-000918-M funded by MCIN/AEI/10.13039/501100011033. We acknowledge financial support from the Generalitat de Catalunya (Grant 2021SGR01095). A.R.-F. acknowledges funding from MIU through Grant No. FPU20/06174.

# A  Limit of static atoms

In this appendix we examine the limit with no tunneling, that is, when $\hat{H} = \hat{H}_{\text{int}}$. In this limit, the Hamiltonian reads

$$\hat{H} = \frac{U_{bb}}{2} \sum_i \hat{n}_{i,b} \left( \hat{n}_{i,b} - 1 \right) + U_{bI} \sum_i \hat{n}_{i,b} \hat{n}_{i,I}. \tag{A.1}$$

In the following, we analyze the solutions of this Hamiltonian for $U_{bI} \geq 0$ by employing mean-field arguments.

## A.1  Ground state

We first examine the ground-state solution of Hamiltonian (A.1). In particular, we examine the distribution of atoms in the lattice and the behavior of the bipolaron energy $E_{bp}$ as defined in Eq. (6). For the latter, we need to compute the energies of the system with zero, one, and two impurities.

For $E_0$ (no impurities), it is easy to see that the ground state of (6) at unity filling simply corresponds to the Mott-like solution of having one boson on each site ($n_{i,b} = 1$). Therefore, $E_0 = 0$ for all interactions.

For $E_1$ (one impurity) we have two scenarios. On one side, for $U_{bI} < U_{bb}$ the bath remains in its Mott-like solution with $n_{i,b} = 1$, so one boson interacts with the single impurity and hence $E_1(U_{bI} < U_{bb}) = U_{bI}$ (miscible phase). On the other side, for $U_{bI} > U_{bb}$ the impurity is able to repel one boson, as it is more energetically favorable to have two bosons at the same site instead of one boson and the impurity. Therefore, $E_1(U_{bI} > U_{bb}) = U_{bb}$. The latter case corresponds to the *phase separation* of the bath and the impurity (immiscible phase).

Finally, for $E_2$ (two impurities) we also have two scenarios. First, for $U_{bI} < U_{bb}/2$ the bath is in its Mott-like state with $n_{i,b} = 1$ while the two impurities occupy arbitrary sites. Therefore, each impurity interacts with one bath's boson and hence $E_2(U_{bI} < U_{bb}/2) = 2U_{bI}$ (miscible phase). In contrast, for $U_{bI} > U_{bb}/2$ the impurities repel the bath strongly enough to induce a bath-impurities phase-separation where the two impurities are in the same site, while the bath occupies the remaining sites (immiscible phase). Therefore, we have two bath's bosons in the same site and hence $E_2(U_{bI} > U_{bb}/2) = U_{bb}$.

With all the energies calculated, the bipolaron energy reads

$$E_{bp} = \begin{cases} 0 & : 0 \leq U_{bI} < U_{bb}/2\,, \\ U_{bb} - 2U_{bI} & : U_{bb}/2 \leq U_{bI} < U_{bb}\,, \\ -U_{bb} & : U_{bb} < U_{bI}\,. \end{cases} \tag{A.2}$$

We show the obtained behavior in Fig. 14. As discussed, for $U_{bI} > U_{bb}/2$ the system undergoes a phase separation where the two impurities occupy the same site, effectively forming a di-impurity dimer. This dimer formation is signaled by the appearance of a negative $E_{bp}$.

As stressed in the figure, for strong repulsion $U_{bI} > U_{bb}$ we have that $E_{bp} = -U_{bb}$, which corresponds to the region where the system with either one or two impurities is immiscible. In contrast, in the region connecting the saturation values ($U_{bb}/2 \leq U_{bI} < U_{bb}$) the system with one impurity is still miscible, while the system with two impurities is already phase-separated.

To illustrate the effect of the tunneling on the bipolaron energy, in Fig. 15 we show $E_{bp}$ for different choices of strong bath's interactions. For $U_{bb}/J_b = 10^3$ the polaron energy is essentially given by its static limit [Eq. (A.2)]. As $U_{bb}$ decreases, the figure shows the crossover of $E_{bp}$ to its behavior where the tunneling becomes important, resulting in a smoother decrease of $E_{bp}$ as reported in the main text [see Fig. 2].

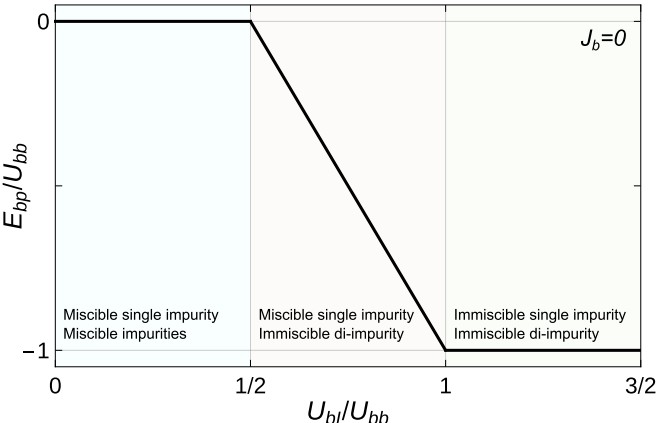

Figure 14: Bipolaron energy $E_{bp}$ as a function of $U_{bI}/U_{bb}$ in the limit of no tunneling as given by Eq. (A.2). The colored regions describe the ground state of the system with one and two impurities.

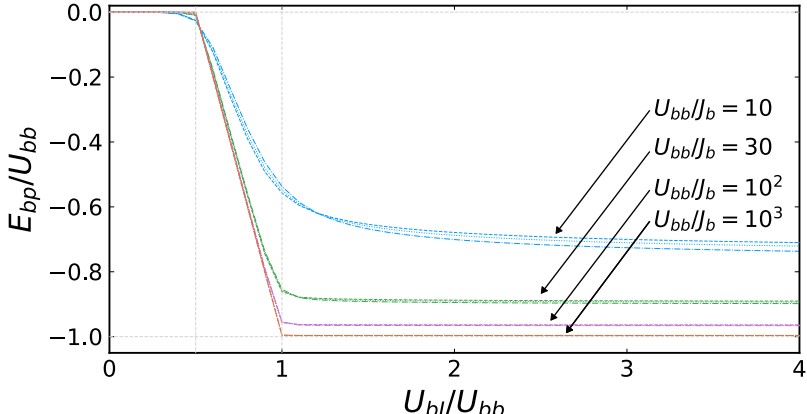

Figure 15: Bipolaron energy $E_{bp}$ as a function of $U_{bI}/U_{bb}$ for large values of $U_{bb}$ (indicated in the legends). The dash-dotted, dotted, and dashed correspond to lattices with $M = 6, 7, 8$ sites, respectively.

It is also worth mentioning that the examined ground state in the static limit has a degeneracy $g_0$ of

$$g_0(J_b = 0) = \begin{cases} \binom{M+1}{2} & : 0 \leq U_{bI} < U_{bb}/2\,, \\ M(M-1) & : U_{bb}/2 < U_{bI}\,. \end{cases} \tag{A.3}$$

This degeneracy is simply a result of the absence of tunneling in the static limit. Indeed, the expression of $g_0$ for $0 \leq U_{bI} < U_{bb}/2$ (miscible phase) corresponds to all the different ways of placing the two impurities in a lattice with $M$ sites. Note that all the ways of placing $N$ bosons in $M$ sites is given by $\mathcal{N}_N^M = \binom{N+M-1}{N}$ [20], and thus $\mathcal{N}_2^M = \binom{M+1}{2}$. Also note that because each site contains one boson of the bath, the bath has only one configuration and is not degenerate. In contrast, the expression for $U_{bb}/2 < U_{bI}$ (immiscible phase) considers that the two phase-separated impurities can be placed in $M$ different sites, leaving $M - 1$ ways to place the site with two bosons from the bath.

### A.2 Low-energy spectrum

We now examine the low-energy spectrum in the limit of static atoms. We examine excited states for lattices with $M = 5$ to better illustrate the energy spectrum. However, larger lattices show an analogous behavior.

Following the discussion for the ground state, the system can show immiscible configurations where the bath and the impurities do not occupy the same sites. In small lattices $M = 5$, some of these immiscible configurations with their respective energies are

$$
\begin{aligned}
&\underline{E_2 = U_{bb}:} &&|21110;00002\rangle. \\
&\underline{E_2 = 2U_{bb}:} &&|22100;00011\rangle, \quad |22100;00002\rangle. \\
&\underline{E_2 = 3U_{bb}:} &&|31100;00011\rangle, \quad |31100;00002\rangle. \\
&\underline{E_2 = 4U_{bb}:} &&|32000;00011\rangle, \quad |32000;0002\rangle.
\end{aligned}
$$

We have used the notation presented in Eq. (4). Note that the first state corresponds to the previously discussed ground state in the immiscible phase. Also, note that we only show one representative state for each configuration, but additional degenerate states are present due to the periodicity of the lattice.

From the examples, it is easy to see that in the lattices examined in this work ($M > 5$), the energy gaps between the first few lowest immiscible states have the same value

$$\Delta E_{\text{imm}} = U_{bb}. \tag{A.4}$$

However, due to the small sizes of the lattices, higher energy states can show larger gaps depending on the number of sites.

On the other hand, miscible (and semi-miscible) configurations have energies that depend on both $U_{bb}$ and $U_{bI}$. Indeed, in the case of $M = 5$, some examples of configurations are

$$
\begin{aligned}
&\underline{E_2 = 2U_{bI}:} &&|11111;11000\rangle, \quad |11111;20000\rangle. \\
&\underline{E_2 = U_{bb} + U_{bI}:} &&|21110;00011\rangle. \\
&\underline{E_2 = U_{bb} + 2U_{bI}:} &&|21110;00110\rangle, \quad |21110;10001\rangle. \\
&\underline{E_2 = U_{bb} + 4U_{bI}:} &&|21110;20000\rangle. \\
&\underline{E_2 = 2U_{bb} + U_{bI}:} &&|22100;00100\rangle. \\
&\underline{E_2 = 2U_{bb} + 2U_{bI}:} &&|22100;10010\rangle, \quad |22100;00200\rangle. \\
&\underline{E_2 = 2U_{bb} + 3U_{bI}:} &&|22100;10100\rangle. \\
&\underline{E_2 = 2U_{bb} + 4U_{bI}:} &&|22100;11000\rangle, \quad |22100;20000\rangle.
\end{aligned}
$$

Further energies with a higher factor of $U_{bb}$ can be obtained by gathering more bath bosons together. These miscible states show energy gaps with different integers of $U_{bb}$ and $U_{bI}$.

We show the analyzed spectrum in Fig. 16. Due to the competition between the $U_{bb}$ and $U_{bI}$, the energy spectrum for small $U_{bI}/U_{bb}$ shows a rich behavior, with many crossings between the different miscible and immiscible states. This is similar to the behavior shown with mobile atoms [see Fig. 7], where for small $U_{bI}$ the spectrum shows many crossings due to the competition between the different parameters in the model.

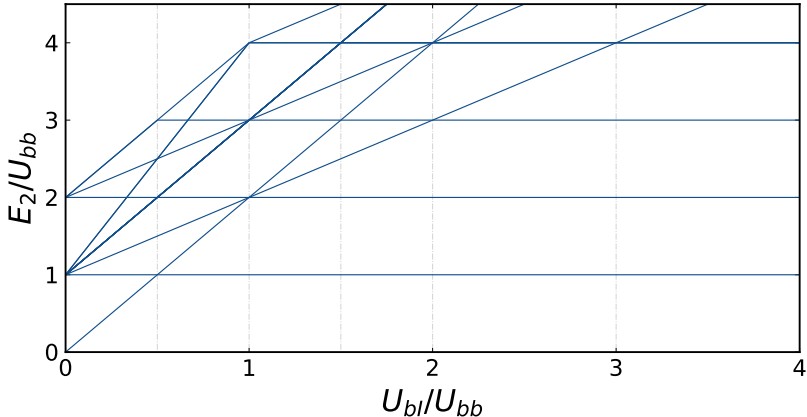

Figure 16: Low-energy spectrum $E_2$ as a function of $U_{bI}/U_{bb}$ for $M = 5$ in the static limit $J_b = 0$.

## B Correlations

To provide a complementary picture of the distribution of the atoms after an interaction quench, in this appendix, we examine the two-body correlator between impurities [49]

$$C_{II}^{(2)}(i) = \langle \Psi | \hat{a}_{i,I}^\dagger \hat{a}_{0,I}^\dagger \hat{a}_{0,I} \hat{a}_{i,I} | \Psi \rangle . \tag{B.1}$$

We present the evolution of $C_{II}^{(2)}$ for interaction quenches in $U_{bI}$ in Fig. 17. We normalize $C_{II}^{(2)}$ by $2/M$ so the maximum value is one. We also again note that in the periodic lattices examined here, the central site 0 is arbitrary.

First, for small interaction quenches (top panels) we observe that the correlations are almost constant with time, showing almost indistinguishable oscillations. As discussed previously, because for such small interaction quenches the dimer should still be formed with similar properties, the system shows minimal oscillations. Panel (a) also shows that $C^{(2)}$ has non-zero values within the three central sites, showing the formation of shallow dimers. In contrast, in panel (b) $C^{(2)}$ is non-zero only at the central site, confirming the formation of a tightly bound dimer in only one site.

On the other hand, for large interaction quenches beyond the critical interaction strength $U_{bI}^*$ examined in Sec. 3.3 (bottom panels), the correlations show the expected large oscillations. Indeed, panels (c) and (d) show that the correlator oscillates between configurations where $C^{(2)}$ shows non-zero values at the farthest sites ($i = \pm 3$) and around the central site. This means that the di-impurity dimer is destroyed and revived, as also shown by the evolution of the average distances $\langle r_{II} \rangle$ [see Fig. 12].

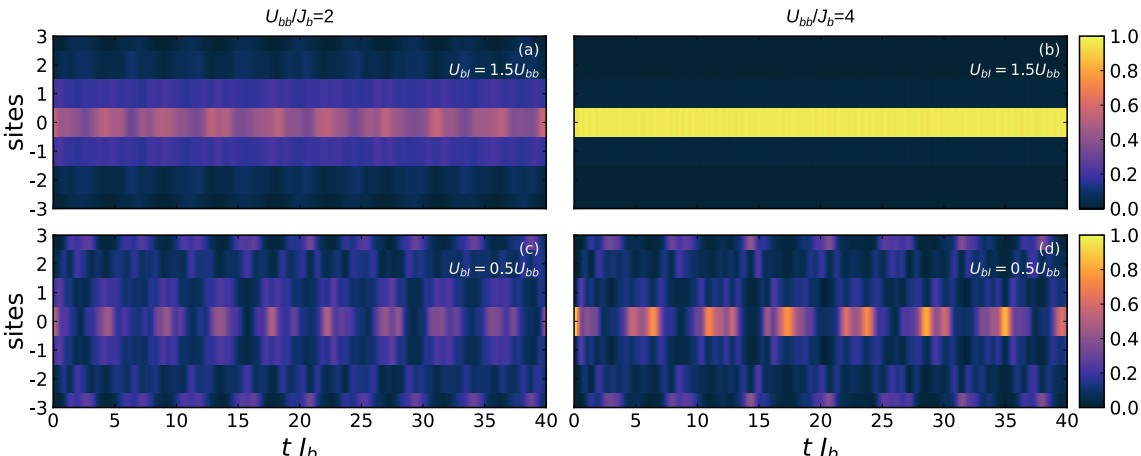

Figure 17: Two-body correlator between impurities $C_{II}^{(2)}$ as a function of time for $M = 7$. The initial state is prepared in the ground state with $U_{bI} = 2U_{bb}$ and either $U_{bb}/J_b = 2$ (left panels) or $U_{bb}/J_b = 8$ (right panels). An interaction quench in $U_{bI}$ is performed at $t = 0$ to the values indicated in the legends.

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
