# Peer review of "Bound impurities in a one-dimensional Bose lattice gas: low-energy properties and quench-induced dynamics"

_SciPost Physics, doi:SciPost Phys. Core 7, 049 (2024)_

## Round 1 · Referee Report · Anonymous (Referee 1) · 2024-3-31

Strengths

  1. The authors study a concrete, experimentally relevant setup with exact diagonalization.
  2. They look at a number of physical quantities to provide a picture for the nature of binding of the two impurities.
  3. They predict collapse-and-revival dynamics of the dimer under an interaction quench, which may motivate experimental observations.

Weaknesses

  1. The study is not particularly novel as the dimer formation and much of the ground-state properties are already known. In particular, there is significant overlap with a paper involving one of the authors (Ref.33).
  2. The study is limited to small system sizes where it is difficult to characterize critical phenomena.
  3. The authors have not discussed how their predictions may be detected experimentally.

Report

The authors study the binding of two spin impurities in a 1D Bose-Hubbard model at unit filling using exact diagonalization. In particular, as the bath-impurity repulsion U_{bI} is increased relative to the bath-bath repulsion U_{bb}, the impurities phase separate from the bath, forming a dimer. Although this binding in the ground state was already known from past work, particularly Ref. [33], the authors provide a more detailed characterization and also analyze low-energy excitations and dynamics following an interaction quench, which they show can lead to collapse and revival of the dimer.

The work is thorough and well written, despite being limited to small systems (9 sites), and constitutes a useful addition to the literature. However, it does not meet the acceptance criteria of SciPost Physics for groundbreaking results or novelty. It could be considered for SciPost Physics Core after addressing the following points.

  1. The authors assume that the two impurity atoms do not interact with each other. Other than simplicity, is there any motivation to work at this limit? Have the authors considered the effect of nonzero interactions?

  2. More generally, it will be useful to discuss in greater depth how realistic the setup is and whether one can observe the results, e.g. collapse and revival, in experiments.

  3. I find the description of Fig. 2(b) in the last paragraph of Sec 3.1 somewhat confusing and lacking physical content. Firstly, the cutoff of 0.4 J_b for weak repulsion is quite arbitrary. (Also, “J_b” is missing in the text.) Secondly, it appears from the figure that the binding energy is quadratic for small for U_{bb} and linear for large U_{bb}, which results in the minima. Can the authors explain the physics behind the quadratic growth in the superfluid region?

  4. On several occasions the authors use the term “critical” interaction strength for dimer formation. However, all of their plots at nonzero tunneling are (necessarily) smooth, becoming sharper in the Mott regime. Is there any reason to expect a phase transition as opposed to a crossover (in the thermodynamic limit)? If not, the authors should qualify their usage.

  5. Can the authors explain why the energy gap in Fig. 6 falls sharply as the impurities become bound? What is the energy scale of dimer tunneling that sets the small gap?

  6. In Fig 7(b) what are the almost equally spaced excitations at large U_{bI}? The gap is much smaller than U_{bb} which would be the cost of multiple doublons in the bath (as in Fig. 16).

  7. In the quench dynamics do the authors expect the collapse and revival to persist for large systems? For instance, how does the corresponding spectral gaps scale with the number of sites M?

  8. Can the authors explain how they get Eq. (9) for the average distance between two free bosons? Is there a similar expression for two free fermions?

Minor points:

  1. In Eq. (7) the one of the occupations should have a subscript “\sigma^{\prime}” as opposed to “\sigma”

  2. The last column in Table I should probably read “r_s^ / r_0” as opposed to “r_s^ / a” since Eq. (10 ) gives r_s^* > r_0.

  3. Below Fig. 4 the authors state that the distance between the impurities vanishes for small U_{bb}. This is only valid if U_{bI} > U_{bb}/2, which would be useful to add here.

  4. In Eq. (A.3) should the degeneracy be M choose 2 for small U_{bI}?

  5. In labeling the curves in Figs. 2(b), 3(b), and 5(b) one should use “U_{bb}” [not “U_{BB}”] like everywhere else.

Requested changes

  1. Discuss experimental realization(s).
  2. Address questions in the report on phase transition and spectral gaps.
  3. Incorporate the minor corrections listed in the report.

  • validity: high
  • significance: good
  • originality: ok
  • clarity: good
  • formatting: excellent
  • grammar: excellent

Author:  Felipe Isaule  on 2024-05-24  [id 4512]

(in reply to Report 1 on 2024-03-31)
Category:
answer to question

We thank the reviewer for the careful reading and detailed evaluation of our manuscript and the positive report. In this revised version, we have rewritten several parts of our manuscript to address the reviewer’s questions and suggestions. Below we respond to each of the reviewer’s comments in turn. We also provide a version of our manuscript with the changes highlighted on:

https://raw.githubusercontent.com/felipeisaule/felipeisaule.github.io/main/files/1D_Lattice_Bose_Bipolarons.pdf

  • “The authors assume that the two impurity atoms do not interact with each other. Other than simplicity, is there any motivation to work at this limit? Have the authors considered the effect of nonzero interactions?”

R: This work is mostly motivated by related studies of the bipolaron problem in BECs, where even if the impurities do not interact among themselves, the bath induces an effective impurity-impurity interaction. We have added a sentence in the paragraph after Eq. (3) stressing this. We have not yet considered the effect of a non-zero impurity-impurity interaction. However, I agree that it is an interesting and also experimentally relevant question. We have added a sentence in the conclusions mentioning this possible future direction.

  • “More generally, it will be useful to discuss in greater depth how realistic the setup is and whether one can observe the results, e.g. collapse and revival, in experiments.”

R: We agree that we did not discuss the experimental realization with enough detail. We separate our answer into the possible experimental realization and the observation of revivals.

On one side, we believe the proposed setup can be achieved in experiment. There are different aspects to consider:

Firstly, and maybe most importantly, the ring configurations considered in this work can be produced in experiments, and indeed have attracted interest over the years to study persistent currents. We have added a sentence in the first paragraph of page 4 stressing this, including new references that discuss the realization of ring configurations.

The employed couplings in our model are mostly within experimental reach, as well as the realization of highly imbalanced mixtures. The last paragraph of page 3 stresses this.

Systems with only a few atoms are more difficult to produce, but they are achievable. This is also stressed in the revised first paragraph of page 4, including a new reference.

Finally, the interaction quenches can be of course realized with Feshbach resonance techniques. We have added a sentence in the first paragraph of Sec. 5 stressing the latter.

Nevertheless, we suspect that some of the features studied in this work will also persist in larger lattices with many atoms, and which could be studied in the future with other numerical techniques. We have added a short comment about this in the last paragraph of the conclusions.

On the other hand, the observation of revivals could be observed with the measurement of the inter-atomic correlations, as done with other ultracold atom mixtures. In the revised manuscript we have commented this after Eq. (18).

In addition, we have added a new paragraph to the conclusions commenting on the possible experimental realization of our model.

  • “I find the description of Fig. 2(b) in the last paragraph of Sec 3.1 somewhat confusing and lacking physical content. Firstly, the cutoff of 0.4 J_b for weak repulsion is quite arbitrary. (Also, “J_b” is missing in the text.) Secondly, it appears from the figure that the binding energy is quadratic for small for U_{bb} and linear for large U_{bb}, which results in the minima. Can the authors explain the physics behind the quadratic growth in the superfluid region?”

R: The separation around 0.4Jb is motivated by the known MF-to-SF transition point in 1D lattices at unity filling (Ubb ≈ 3.6Jb). While such a transition point corresponds of course to the limit of an infinite lattice, we find that in our small few-body system the behavior of the system does change around that region, and it is thus a good reference point to separate the regime of weak and strong Ubb. It is true that this does not correspond to a well-defined transition point, but it is not a completely arbitrary point either. However, we agree that this was not well-explained in our original manuscript. In particular, we did not explain that it just corresponds to a reference interaction. The mentioned paragraph of Sec. 3.1 has been revised (and separated into two paragraphs) to better explain this separation. The missing Jb has also been fixed. In addition, several mentions of this separation point 0.4Jb have been removed throughout the text and replaced with better explanations. Concerning the minimum seen in Fig 2b, while it is a curious behavior, it is in part an artifact of the scaling, and it is not a real energy minimum. We have changed the panel (b) of Fig. 2 to show Ebp scaled by Jb which does not distort the curves. The inset still shows Ebp/Ubb to show the behavior for large Ubb. We have changed the caption and text accordingly.

  • “On several occasions the authors use the term “critical” interaction strength for dimer formation. However, all of their plots at nonzero tunneling are (necessarily) smooth, becoming sharper in the Mott regime. Is there any reason to expect a phase transition as opposed to a crossover (in the thermodynamic limit)? If not, the authors should qualify their usage.”

R: We agree that we observe crossovers instead of well-defined transitions with critical points, and thus some of our remarks were not entirely accurate. In the revised manuscript, this crossover is explained in much more detail in several parts, particularly in the paragraph starting at the end of page 5 and in the first paragraph of Sec. 3.3. Moreover, in the revised manuscript we use the terms transitions and critical interaction more sparsely and accurately. Nevertheless, the tunneling correlator Ct (Sec. 3.3) gives us a somewhat well-defined critical point, although this is not necessarily the point for phase separation. We have explained these subtleties in much more detail in the revised manuscript.

  • “Can the authors explain why the energy gap in Fig. 6 falls sharply as the impurities become bound? What is the energy scale of dimer tunneling that sets the small gap?”

R: Following the last question, for large Ubb there is a somewhat sharp crossover around the point with Ct=0 from a miscible to phase-separated configuration (explained in more detail in the revised section 3). In addition, in the non-interacting limit UbI=0 (miscible phase), the ground state is non-degenerate. In contrast, for infinite UbI (phase separation) the ground state is (almost) degenerate, resulting in a vanishing energy gap. Therefore, it can be expected that, particularly for large Ubb, the energy gap decreases sharply around UbI~Ubb/2. In the revised manuscript, the last three paragraphs of Sec. 4.1 have been rewritten with a more detailed discussion of these subtleties.

  • “In Fig 7(b) what are the almost equally spaced excitations at large U_{bI}? The gap is much smaller than U_{bb} which would be the cost of multiple doublons in the bath (as in Fig. 16).”

R: They correspond to excitations of a bath of M bosons in an M-1 lattice due to the phase separation between the bath and the impurity (which occupies the remaining site). Indeed, we have performed calculations for a one-component Bose-Hubbard model with M bosons in M-1 sites and we obtain the same low energy gaps as for large UbI. While this was explained in the previous version, the revised Sec. 4.2 explains this in more detail, particularly in the paragraph starting at the end of page 12.

  • “In the quench dynamics do the authors expect the collapse and revival to persist for large systems? For instance, how does the corresponding spectral gaps scale with the number of sites M?”

R: The collapses and revivals are indeed also present in larger lattices. In fact, within the lattices examined in this work, we have found that the only difference between lattices with different M is the periods of oscillations. We observe that the periods increase with the number of sites. However, because the spectrum and periods depend on M, it is difficult to compare the results with different lattice sizes. Therefore, we have not been able to find exactly how the oscillations scale with M, and thus it is difficult to make predictions for very large lattices. Nevertheless, in the ring setups that we propose, which could be achieved in experiments, we believe the revivals should be present. We have added a few comments before 5.1 and a new paragraph at the end of Sec. 5 stressing this.

  • “Can the authors explain how they get Eq. (9) for the average distance between two free bosons? Is there a similar expression for two free fermions?”

R: In a sense it is a problem of combinatorics. Basically, one needs to consider all the possible ways to place two particles in M sites. Because the particles do not interact, each of these possible configurations is equally probable. Then one needs to associate a distance to each configuration (given by subtraction of the site's indexes) and take the average. We have checked that a numerical calculation of two free bosons gives the exact same result. It can also be checked by hand for small lattices. We have expanded the paragraph after Eq. (9) explaining this. Concerning fermions, we have not been able to come up with an analogous formula, so we calculate that distance numerically [used in Eq. (10)] . Nevertheless, for sure such a formula exists in the literature, but we have not been able to find it.

Minor points:

  • “In Eq. (7) the one of the occupations should have a subscript “\sigma^{\prime}” as opposed to “\sigma” “

R: Fixed. Thank you.

  • “The last column in Table I should probably read “r_s^ / r_0” as opposed to “r_s^ / a” since Eq. (10 ) gives r_s^* > r_0.”

R: Yes, that is correct. It should read r_s^* / r_0. Fixed. Thank you.

  • “Below Fig. 4 the authors state that the distance between the impurities vanishes for small U_{bb}. This is only valid if U_{bI} > U_{bb}/2, which would be useful to add here.”

R: Thank you for suggesting this. Added.

  • “In Eq. (A.3) should the degeneracy be M choose 2 for small U_{bI}?”

R: The expression in (A.3) is correct, but it was very badly explained. The paragraph after (A.3) has been revised in the new manuscript to better explain the degeneracy.

  • “In labeling the curves in Figs. 2(b), 3(b), and 5(b) one should use “U_{bb}” [not “U_{BB}”] like everywhere else.”

R: Fixed. Thank you.

“Requested changes 1. Discuss experimental realization(s). 2. Address questions in the report on phase transition and spectral gaps. 3. Incorporate the minor corrections listed in the report.”

R: We believe we made all the requested changes in the revised manuscript.

---

## Round 1 · Referee Report · Anonymous (Referee 2) · 2024-5-1

Strengths

1) Comprehensive description 2) Detailed characterization of microscopic properties 3) Exact calculations 4) Indications of persistent behavior from small to larger systems

Weaknesses

1) Lack of explanation for certain processes 2) Some of the presented results, such as the formation of impurity dimers, have also been reported elsewhere

Report

In the present work, the authors study the stationary properties and the interaction quench dynamics of two bosonic impurities in a few-body bosonic bath. Both species are trapped in an one-dimensional lattice. The situation of unit filling is considered throughout and exact diagonalization is employed to perform the calculations. Regarding the ground state, the formation of a bound di-impurity dimer caused by the genuine phase-separation process in two-component settings is identified and characterized in detail in terms of the system parameters. Furthermore, following interaction quenches from strong to weak impurity-bath repulsions revealed the regions where the aforementioned dimer persists or can be dynamically destroyed indicating an orthogonality catastrophe.

The results are interesting providing insights into the formation of impurity bound states immersed in bosonic baths under the influence of a lattice geometry. I believe that they will be proven useful for future investigations on impurity and in general quasiparticle states especially concerning their non equilibrium dynamics. This topic is also of interest from an experimental point of view. However, I have several comments and questions regarding the presentation and more importantly the interpretation of the results that should be addressed before final recommendation. A list of suggestions for improvement follows.

Requested changes

1) On page 4, it is stated that “most of the parameters considered here lie around the one-dimensional superfluid-to-insulator transition”. Does this statement refers solely to the majority component? Please clarify.

2) Before Equation (4) the word “stated” should read “state”

3) Equation (5) is the total wave function if I understand correctly and not a general wavevector.

4) Regarding the numerical ED method used, I wonder whether only the lowest band states of the lattice are taken into account or also energetically higher ones in the current study. In this context, what is the relation between the first excited state discussed in section 4 and higher-band states of the lattice? Please clarify. Again here, how numerical convergence is judged in the present work?

5) At the beginning of Section 3, the concept of “di-impurity bound states” is mentioned but not explained. Please provide a brief explanation since it has been studied also elsewhere.

6) In Figure 2 the dashed and dashed-dotted lines and especially the vertical dashed-dotted lines are hardly visible. I suggest to increase their fonts. The same holds for Figures 3, 5, 6, 7 and 8.

7) What is the phase-separation condition in the present setting in terms of the interactions of the bath atoms and the interaction between the impurities and the bath atoms?

8) On page 5, I do not fully understand the comment about the bipolar energy which shows an an abrupt decrease at a fixed ratio and for smaller values a more continuous crossover. I see in all cases a rather continuous curve. Please explain better what is meant here.

9) What is the origin of the energy minimum observed in Figure 2(b) at strong impurity bath interactions? Is it the phase separation?

10) In Equation (7) the index \alpha is referred to as the lattice spacing while in Equation (5) it is the state. This is a bit confusing. I suggest to modify one of these notations.

11) It does not become clear why and how the competition between phase-separation and Mott-insulator or superfluid character of the bath impacts the impurities bound state. It this related to the compressibility of the bath? Please clarify.

12) An interesting possibility would be instead of examining the first excited state of the system to consider only the impurity(ies) in their excited states while the bath in its lowest state. Would then the di-impurity bound state exist? If yes, how its properties will be changed.

13) Are the oscillations observed in the overlap being present for larger systems (in particular lattice sizes and bath atom numbers) or do they become more prominent for smaller system sizes? Is it possible to discern finite effects from genuine ones here? I suspect that there are indeed genuine effects since phase-separation and Mott-insulator to superfluid transition play a decisive role in the behavior of the overlap and in the dynamics of the system. Please clarify.

14) In all dynamical cases considered, if I understand correctly, tunneling of the impurities is prevented due to the large energy gap. What happens for interactions/hoping coefficients where tunneling is favored? Is it possible that the di-impurity tunnels an entity and thus two-body effects are prominent?

Recommendation

Ask for minor revision

  • validity: high
  • significance: high
  • originality: ok
  • clarity: high
  • formatting: good
  • grammar: good

Author:  Felipe Isaule  on 2024-05-24  [id 4511]

(in reply to Report 2 on 2024-05-01)
Category:
answer to question

We thank the reviewer for the careful reading and detailed evaluation of our manuscript and for finding our work interesting. In this revised version, we have rewritten several parts of our manuscript to address the reviewer’s questions and suggestions. Below we respond to each of the referee’s comments in turn. We also provide a version of our manuscript with the changes highlighted on:

https://raw.githubusercontent.com/felipeisaule/felipeisaule.github.io/main/files/1D_Lattice_Bose_Bipolarons.pdf

1) “On page 4, it is stated that “most of the parameters considered here lie around the one-dimensional superfluid-to-insulator transition”. Does this statement refer solely to the majority component? Please clarify.”

R: Yes, it refers to the bath. Thanks. We have revised that sentence in the new manuscript.

2) “Before Equation (4) the word “stated” should read “state” ”

R: Fixed. Thank you.

3) “Equation (5) is the total wave function if I understand correctly and not a general wavevector.”

R: Yes, fixed. Thank you.

4) “Regarding the numerical ED method used, I wonder whether only the lowest band states of the lattice are taken into account or also energetically higher ones in the current study. In this context, what is the relation between the first excited state discussed in section 4 and higher-band states of the lattice? Please clarify. Again here, how numerical convergence is judged in the present work?”

R: Hubbard models assume that all particles are in the lowest Bloch band, and thus our model does not consider higher-band states. We have added a short comment before Eq. (1) stressing this. We employ the standard and widely-used ARPACK package to perform the diagonalizations. The convergence is easy to control by adjusting the parameters of the Lanczos algorithm, starting the iteration with a solution from a previous diagonalization, and, of course, by examining the error provided by the subroutine. We have added a sentence in the paragraph after Eq. (5) stressing this.

5) “At the beginning of Section 3, the concept of “di-impurity bound states” is mentioned but not explained. Please provide a brief explanation since it has been studied also elsewhere.”

R: In the revised manuscript the beginning of Sec. 3 has been expanded explaining this point in more detail. Related to this question, several parts of the manuscript (particularly Sec. 3) have been revised to better explain the di-impurity states.

6) “In Figure 2 the dashed and dashed-dotted lines and especially the vertical dashed-dotted lines are hardly visible. I suggest to increase their fonts. The same holds for Figures 3, 5, 6, 7 and 8.”

R: Thanks for the suggestion. The figures have been updated with thicker lines. In addition, the vertical and horizontal lines now use a darker gray to highlight them more.

7) “What is the phase-separation condition in the present setting in terms of the interactions of the bath atoms and the interaction between the impurities and the bath atoms?”

R: In the static limit, the phase separation occurs exactly at UbI=Ubb/2 [Appendix A.1], which can be understood from mean-field arguments. In the mobile case (main text), there is no well-defined transition, as the system shows instead a crossover between the miscible and phase-separated configuration. However, this crossover still occurs around UbI=Ubb/2 as suggested by several figures in our manuscript. We have added a new paragraph starting at the end of page 5 explaining this in much more detail. The crossover and the condition for phase separation are also stressed in more detail in the revised Sec. 3.2. In particular, the phase separation is mainly conveyed by our results for the average distances (illustrated by Fig. 4).

8) “On page 5, I do not fully understand the comment about the bipolar energy which shows an abrupt decrease at a fixed ratio and for smaller values a more continuous crossover. I see in all cases a rather continuous curve. Please explain better what is meant here.”

R: Yes, we agree that this comment was not accurate. Connected to the last question, what we tried to say is that for small Ubb the bipolaron energy shows a very smooth crossover from a miscible to an immiscible phase, while for large Ubb the crossover is more abrupt (but still a crossover). In the added paragraph to Sec. 3.1 this crossover behavior is explained more accurately. This crossover is also explained in more detail in the revised sections 3.2 and 3.3

9) “What is the origin of the energy minimum observed in Figure 2(b) at strong impurity bath interactions? Is it the phase separation?”

R: While the minimum is a curious behavior, it is in part an artifact of the scaling, and it is not a real energy minimum. We have changed the panel (b) of Fig. 2 to show Ebp scaled by Jb which does not distort the curves. The inset still shows Ebp/Ubb to show the behavior for large Ubb. We have changed the caption and text accordingly.

10) “In Equation (7) the index \alpha is referred to as the lattice spacing while in Equation (5) it is the state. This is a bit confusing. I suggest to modify one of these notations.”

R: The lattice spacing was denoted by “a”, not by “alpha”, but we agree that it is too difficult to tell apart. We have changed the notation of the lattice spacing to “d” throughout the text. Thank you.

11) “It does not become clear why and how the competition between phase-separation and Mott-insulator or superfluid character of the bath impacts the impurities bound state. Is this related to the compressibility of the bath? Please clarify.”

R: Yes, it is due to the compressibility. If the bath is in a Mott-like state, due to the phase separation the bosons of the bath occupy M-1 sites (M the number of sites) and cannot be moved, and thus the two impurities must occupy the same remaining site. This is illustrated in Fig. 4b. In contrast, with a superfluid bath, while there is a phase separation and the impurities form a dimer, the impurities can compress the bath. This produces a dimer with a larger size. This is illustrated in Fig. 4a. Because this was not entirely clear, we have rewritten the second paragraph of page 8 explaining this in more detail.

12) “An interesting possibility would be instead of examining the first excited state of the system to consider only the impurity(ies) in their excited states while the bath in its lowest state. Would then the di-impurity bound state exist? If yes, how its properties will be changed.”

R: Yes, that is an interesting future direction. The di-impurity states continue to exist in the lower part of the spectrum. As we discuss in Sec. 4.2 (page 12), in limiting interaction scenarios we can understand certain energy excitations simply as excitations of a bath confined in an M-1 lattice. This is indeed due to the phase separation and the formation of a dimer in the remaining site. We expect that the average sizes of the dimers will change due to the excitation of the atoms, but of course, that should be investigated. Moreover, some excited states can correspond to circular currents, which could give rise to an interesting interplay between the bath and the impurity. The revised Sec. 4.2 (mainly the paragraph beginning at the end of page 12) discusses the phase separation in the low-energy spectrum in much more detail. Moreover, we have added the study of excited states to the list of future directions at the end of the conclusions.

13) “Are the oscillations observed in the overlap being present for larger systems (in particular lattice sizes and bath atom numbers) or do they become more prominent for smaller system sizes? Is it possible to discern finite effects from genuine ones here? I suspect that there are indeed genuine effects since phase-separation and Mott-insulator to superfluid transition play a decisive role in the behavior of the overlap and in the dynamics of the system. Please clarify.”

R: Yes, we have found that lattices with other sizes also show the same kind of strong oscillations. The only relevant difference between lattices with different sizes is that the periods of oscillations increase with the number of sites. This is because the energy gaps in the spectrum depend on the lattice size. Therefore, differences in the periods make comparisons between different lattice sizes difficult, which is the reason why we only show results for M=7 in Sec. 5. We have added a few comments before 5.1 and a new paragraph at the end of Sec. 5 stressing this.

14) “In all dynamical cases considered, if I understand correctly, tunneling of the impurities is prevented due to the large energy gap. What happens for interactions/hopping coefficients where tunneling is favored? Is it possible that the di-impurity tunnels an entity and thus two-body effects are prominent?”

R: No, the impurities can indeed tunnel. Only if the tunneling parameter is negligible comparable to the interaction strengths, then the tunneling is not favored. This occurs when the atoms are too massive or when the lattice wells have a large depth. In fact, in Sec. 3.3 we show that one signature of the formation of di-impurity dimers is that they can tunnel together as a bath pair. The energy gaps are related instead to the periods of oscillation, as explained in Sec. 5.2.

---

## Round 2 · Referee Report · Anonymous (Referee 2) · 2024-5-30

Strengths

1) Comprehensive description
2) Detailed characterization of microscopic properties
3) Exact calculations
4) Indications of persistent behavior from small to larger systems

Weaknesses

1) Some of the presented results, such as the formation of impurity dimers, have also been reported elsewhere

Report

The authors addressed the comments in response to the previous referee report. As such, the revised manuscript is significantly improved in several aspects. Indeed, the description and interpretation of the results are now more solid. For instance, the behavior of the bipolaron energy and its transition behavior is clear, the competition between interspecies phase-separation and Mott-insulator of the bath is better understandable, the role and properties of the di-impurity are better conveyed and the method description is improved. Such systems, despite being few-body, are valuable for our understanding on the microscopic processes and are certainly accessible experimentally nowadays. For these reasons, I recommend this work for publication in Sci Post Physics.

Recommendation

Publish (meets expectations and criteria for this Journal)

---

## Round 2 · Referee Report · Anonymous (Referee 1) · 2024-6-13

Strengths

Comprehensive and detailed study
Well formed narrative that provides a clear physical picture
Experimentally relevant predictions

Weaknesses

Not much new physics apart from the collapse and revival dynamics
Limited to small system sizes

Report

I thank the authors for adequately addressing all of the technical points in the previous report. However, as before, I think the work is better suited for SciPost Physics Core. The main physical phenomenon the work centers around—that of dimer formation via bath-mediated interaction—was already known from past work, which also explored interaction quenches [49, 44]. I agree that the present work is more comprehensive and perhaps more physically illuminating, but I wouldn’t say that it opens a new pathway for research. Thus, it does not meet the criteria for SciPost Physics for me. As I see it, the really new finding here is the collapse and revival dynamics of the dimer for small systems, which would be interesting to observe experimentally. However, I think this is more appropriate for SciPost Physics Core.

Recommendation

Accept in alternative Journal (see Report)

---

## Round 2 · Author Response

Dear Editor,

We thank the reviewers for the detailed evaluation of our manuscript. We have revised several parts of our manuscript, taking into account all questions and suggestions of the reviewers. We believe these changes have greatly improved the quality of our work.

While it is true that our work shares an overlap with a previous article, this partial overlap only applies to Secs. 3.1 and 3.2, where we summarized some previous results to present a cohesive manuscript. In the current revision, we have provided details on the possible experimental realization of our model and emphasized the novelty of our results. We believe our work includes enough new and relevant results to be published in Scipost Physics. However, should the Editorial College and reviewers consider that Scipost Physics Core is the most appropriate journal for our manuscript, we would accept it to reduce the delay in getting our work published.

Please find below a list of the changes made. We also provide a version of our manuscript with the changes highlighted on:
https://raw.githubusercontent.com/felipeisaule/felipeisaule.github.io/main/files/1D_Lattice_Bose_Bipolarons.pdf

Additionally, we provide complete responses to each report on the original submission page.

Thank you for receiving our manuscript, and we hope that it is now acceptable for publication.

Yours sincerely,

Dr. Felipe Isaule
Corresponding author

---

## Round 2 · List of Changes

*Updated published preprints.
*Added some references, mostly to justify some additions to the new manuscript:
-New Journal of Physics 25, 093032 (2023)
-Nature Physics 17 (6), 731-735 (2021)
-Physical Review A 105, L021303 (2022)
-New Journal of Physics 21, 053024 (2019)
-Physical Review B 99, 205414 (2019)
-Physical Review Letters 95, 063201 (2005)
-New Journal of Physics 11, 043030 (2009)
-Reviews of Modern Physics 82, 1225 (2010)
-Reports on Progress in Physics 75, 046401 (2012).
*A sentence in the abstract was rewritten to avoid referring to critical points, as suggested by the reviewers.
*A few words have been rewritten in the last paragraphs of the introduction.
*Several parts of Sec. 2 have been rewritten to explain questions raised by the reviewers.
*The first paragraph of Sec. 3 has been rewritten to better explain the di-impurity states.
*The last three paragraphs of Sec. 3.1 have been rewritten to better explain some aspects of our results.
*Added a sentence after Eq. (9) explaining how to obtain the formula for r0.
*A sentence before Table 1 has been rewritten to highlight the crossover.
*A typo in Table 1 has been fixed.
*The second paragraph of page 8 has been rewritten to better explain the phase separation.
*The first paragraph of Sec. 3.3 has been rewritten to better explain the crossover and the tunneling correlator Ct.
*The last three paragraphs of Sec. 4.1 have been rewritten to better explain the energy gaps.
*The fourth and fifth paragraphs of Sec. 4.2 have been rewritten to better explain the excited states.
*The first paragraphs of Sec. 5 (before 5.1) have been rewritten to explain the possible experimental observation of revivals and to justify the choice for the number of sites.
*Selected sentences of Sec. 5.1 have been rewritten to highlight the crossover behavior of our model.
*A new paragraph has been added at the end of Sec. 5 to explain the dependence of our results on the number of sites.
*The second paragraph of the conclusions has been rewritten to better explain the formation of dimers.
*A new paragraph has been added to the conclusions to highlight the possible experimental realization of our model.
*The last paragraph of the conclusions has been revised to include additional future directions for our work.
*The paragraph after (A.3) has been rewritten to better explain the degeneracy of the ground state.
*The gray lines in the figures have been updated with thicker and darker lines.
*The notation of the lattice’s spacing was changed from “a” to “d”.
*A typo in Table 1 was fixed.

---

## Editorial Decision

published